# Kinetics of initiating polypeptide elongation in an IRES-dependent system

**Haibo Zhang[†], Martin Y Ng, Yuanwei Chen[‡], Barry S Cooperman***

Department of Chemistry, University of Pennsylvania, Philadelphia, United States

**Abstract** The intergenic IRES of Cricket Paralysis Virus (CrPV-IRES) forms a tight complex with 80S ribosomes capable of initiating the cell-free synthesis of complete proteins in the absence of initiation factors. Such synthesis raises the question of what effect the necessary IRES dissociation from the tRNA binding sites, and ultimately from all of the ribosome, has on the rates of initial peptide elongation steps as nascent peptide is formed. Here we report the first results measuring rates of reaction for the initial cycles of IRES-dependent elongation. Our results demonstrate that 1) the first two cycles of elongation proceed much more slowly than subsequent cycles, 2) these reduced rates arise from slow pseudo-translocation and translocation steps, and 3) the retarding effect of ribosome-bound IRES on protein synthesis is largely overcome following translocation of tripeptidyl-tRNA. Our results also provide a straightforward approach to detailed mechanistic characterization of many aspects of eukaryotic polypeptide elongation.

**\*For correspondence:**
cooprman@pobox.upenn.edu

**Present address:** [†]Spark Therapeutics, Philadelphia, United States; [‡]Hacker Way, Menlo Park, United Stetas

**Competing interests:** The authors declare that no competing interests exist.

## Introduction

Initiation of protein synthesis in eukaryotic cells proceeds via two well-established pathways. The cap-dependent pathway involves recognition of 7-methyl-guanosine at the 5'-terminus of mRNA by a preinitiation complex of 40S ribosomal subunit and a host of initiation factors prior to a scanning step that results in initiator aminoacyl-tRNA(aa-tRNA) pairing with a cognate start codon, followed by 60S binding to form the 80S initiator complex (*Jackson et al., 2010*; *Aitken and Lorsch, 2012*). The second pathway involves binding of the ribosome to an internal ribosome entry site (IRES), a structure that is present in many virus-encoded mRNAs, as well as in some cellular mRNAs (*Fitzgerald and Semmler, 2009*). Initiation of protein synthesis from an 80S·IRES complex can take place in the absence of some or even all of the initiation factors required in the cap-dependent pathway (*Filbin and Kieft, 2009*), depending on the IRES source. The intergenic IRES of Cricket Paralysis Virus (CrPV-IRES) forms a complex with 80S ribosomes that is capable of initiating the synthesis of complete proteins in cell-free assays completely lacking initiation factors (*Jan et al. 2003*; *Pestova and Hillen, 2003*). More recently, high resolution structural studieshave shown that, prior to polypeptide chain initiation, the closely related Dicistroviridae IRES structures from CrPV (*Fernandez et al., 2014*; *Muhs et al., 2015*) and Taura syndrome virus (*Koh et al., 2014*) occupy all three tRNA binding sites (E, P, and A) on the ribosome, with the protein coding region beginning immediately downstream from IRES segment occupying the A-site (*Figure 1*).

CrPV-IRES binds with high affinity ($K_d \sim 10$ nM) to the 80S ribosome (*Jang and Jan, 2010*), raising the question of what effect the necessary IRES dissociation from the tRNA binding sites, and ultimately from all of the ribosome as well, has on the rates of initial peptide elongation steps as nascent peptide is formed (*Muhs et al., 2015*). Since prior to the work reported in this paper nothing had been published concerning the rate of initial oligopeptide synthesis by an 80S·CrPV-IRES complex, it has been unclear whether there is a retarding effect due to the presence of IRES on the ribosome, and, if so, how many cycles of peptide elongation are required before the ribosome begins to form peptide bonds at a higher rate. In considering this question, we make use of the

**eLife digest** Inside cells, machines called ribosomes make proteins using instructions carried by molecules of messenger RNA (or mRNA). The ribosomes bind to the mRNA and then move along it to assemble the proteins in a process called translation. The first step of translation – when the ribosome binds to the mRNA – is known as initiation. In human and other eukaryotic cells, initiation mainly occurs through a mechanism that requires many proteins called initiation factors to recruit the ribosome to a cap structure formed at one end of the mRNA.

When viruses infect cells, they hijack the ribosomes of the host cell to produce large quantities of viral proteins. However, unlike their host cells, many viruses use a different pathway to initiate translation of their mRNAs. The mRNAs of these viruses have regions known an internal ribosome entry sites (IRESs) that host cell ribosomes can bind to instead.

After initiation, the ribosome progressively assembles the building blocks of proteins (amino acids) into a chain until the new protein is complete. Molecules called transfer RNAs bind to individual amino acids and bring them to the ribosome. Previous research has shown that, prior to initiation, IRESs on Cricket Paralysis Virus mRNAs bind to the ribosome and occupy sites where transfer RNAs would normally bind. However, it was not clear how this affects the elongation process. Zhang et al. now address this question using a cell-free system that allowed them to recreate and observe translation outside of the normal cell environment.

Zhang et al. found that the binding of an IRES to a ribosome slows down the early steps of elongation. A likely explanation for this is that the IRES elements have to be displaced from the ribosome before the incoming transfer RNAs can occupy the three tRNA sites. However, as elongation progresses, the effects of the IRES elements are overcome and the pace of elongation increases significantly. Zhang et al.'s findings provide a convenient approach that could be used for future studies of elongation. This approach could also help researchers find out how abnormalities in translation contribute to human diseases, including muscle-wasting disorders.

simplified 12-step scheme of initial tetrapeptide synthesis shown in *Figure 2*, which provides a useful framework for presenting the results described in this paper. In this scheme Steps 1–3 show the reactions required for initial binding of the first tRNA to the A site followed by translocation to the P-site, and reactions 4–6, 7–9, and 10–12 represent three elongation cycles, ending with P-site bound tetrapeptidyl-tRNA, completing the third cycle of polypeptide synthesis. This model makes the reasonable assumption that binding of successive aminoacyl-tRNAs (aa-tRNAs) cognate to the mRNA requires the progressive removal of IRES structures from each of the tRNA binding sites, such that translocation of dipeptidyl tRNA to the P-site (structure **7**) requires removal of the IRES from the last of the three tRNA binding sites. In the work reported below, we demonstrate first, that the initial elongation steps are indeed quite slow and are limited by the translocation step of the elongation cycle, and second, that the rate of elongation accelerates following translocation of tripeptidyl-tRNA to the P-site.

## Results

In our experiments, eukaryotic ribosomes are prepared from shrimp cysts (*Iwasaki and Kaziro, 1979*), elongation factors are prepared from yeast, and charged tRNAs are prepared from yeast and *E. coli*. In addition, the peptide coding sequence attached to the 3'-end of the CrPV-IRES (*Figure 1*) has been mutated for ease in detection of peptide synthesis via $^{35}$S-Met incorporation. In all such mutants the initial wt-codon triplet GCU encoding Ala has been replaced by UUC, encoding Phe, a change that has little effect on the expression of active luciferase in a cell-free protein synthesis assay (*Figure 1—figure supplement 1*). The initial coding sequences of the mutants used in this work are presented in *Supplementary file 1*. Collectively, they allow monitoring of the rates of PheMet, PheLysMet, PheValLysMet and PheLyVaIArgGlnTrpLeuMet synthesis. In presenting the results below, Steps 1–12 and structures **1 – 13** are as described in the scheme for initial tetrapeptide synthesis proposed in *Figure 2*. Values of $t_{1/2}$ for Steps 1–12, determined as described, are summarized in *Table 2*.

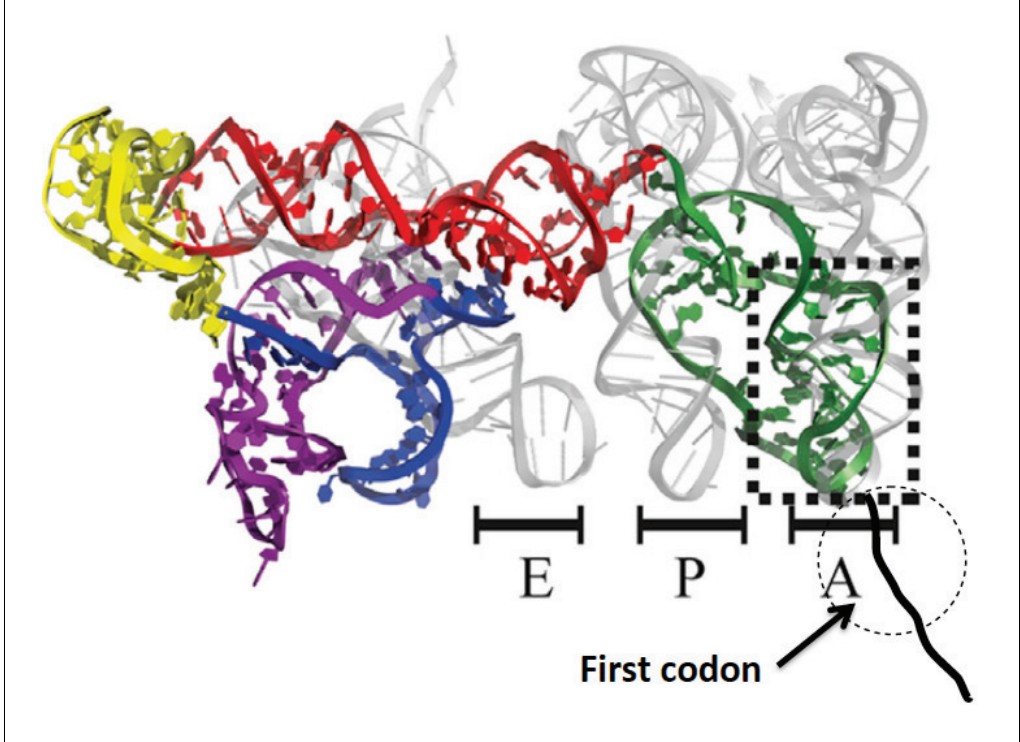

**Figure 1.** Structure of CrPV-IRES bound to the 80S ribosome superposed on A, P, and E tRNA binding sites. The position of the first codon is indicated. Adapted from *Fernandez et al. (2014)*.
The following figure supplement is available for figure 1:

**Figure supplement 1.** In vitro translation of firefly luciferase with WT and mutated F-IRES mRNA.

## Rates of Phe-TC binding to the 80S·IRES complex: Steps 1–3, structures 1–4

We previously have utilized two assays to measure binding of the ternary complex Phe-tRNA$^{Phe}$·eEF1A·GTP (Phe-TC) to the 80S·CrPV-IRES (80S·IRES) complex (*Ruehle et al., 2015*). The increase in proflavin-labeled Phe-tRNA$^{Phe}$ fluorescence anisotropy measures binding to either the A- or P-site (structures **3** and **4**, respectively, *Figure 2*). [$^{3}$H]-Phe-tRNA$^{Phe}$ cosedimentation with the 80S·IRES complex measures accumulation of **4** only, since A-site binding is too labile to survive the ultracentrifugation step (*Yamamoto et al., 2007*).

In *Figure 3* we present time-resolved application of the anisotropy assay that allows us to measure the rates of Phe-TC binding to form Structure **3** from **1**. These resultswere fit to the scheme shown in *Figure 2*, giving values for $k_1$, $k_{-1}$, and $k_2$ in both the presence and absence of eEF2·GTP that are summarized in *Table 1*. In the absence of eEF2 (blue trace), the equilibrium position of Step 1, a so-called pseudo-translocation step (*Muhs et al., 2015*) in which the IRES vacates the A-site, favors Structure **1** over Structure **2** by approximately 20-fold, consistent with recent structural studies (*Fernandez et al., 2014*; *Koh et al., 2014*; *Muhs et al., 2015*). Phe-TC binds to Structure **2** yielding Structure **3**, in a process where the rate-limiting step is the conversion of Structure **1** to Structure **2**. Preincubation of 80S·IRES complex with 1 µM or 3 µM eEF2·GTP leads to clear biphasic binding of Phe-TC, with the more rapid and slower phases each accounting for ~50% of binding, respectively (red and black traces). These results indicate that, consistent with recent results of *Petrov et al. (2016)*, the equilibrium between Structures **1** and **2** is shifted in the presence of eEF2·GTP, such that approximately half of 80S·IRES is present as **2**.Phe-TC binding to **2**, resulting in the formation of Structure **3**, accounts for the rapid phase in the red and black traces. Further formation of **3** is limited by the slower rate of **1** to **2** conversion. Although added eEF2·GTP decreases all three apparent rate constants, the effect is much greater on $k_{-1}$ (~50-fold reduction) than on either $k_1$ (~twofold

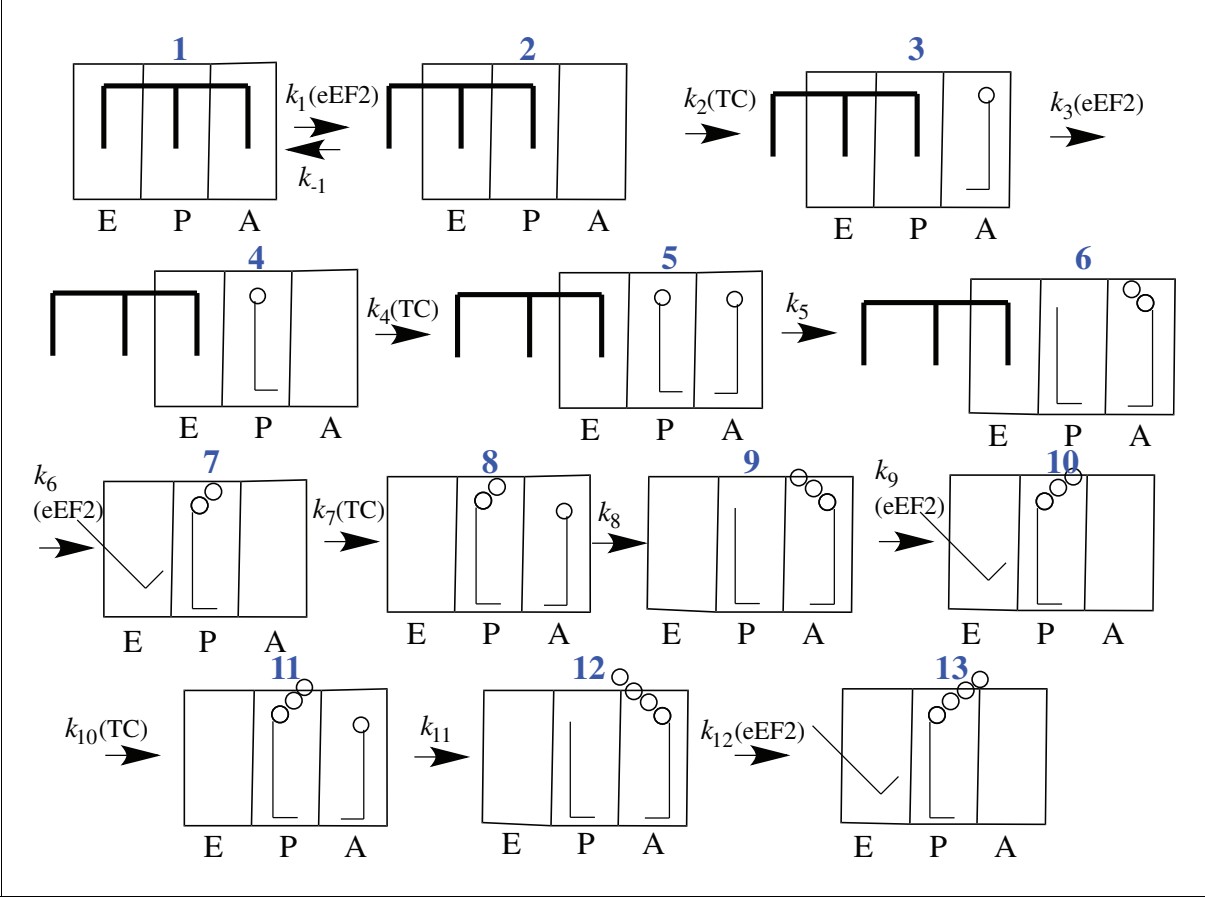

**Figure 2.** Proposed scheme for initial tetrapeptide synthesis on CrPV IRES-programmed ribosomes. This simplified scheme neglects the several substeps, including GTP hydrolysis, $P_i$ release, and elongation factor release, that accompany both productive binding of ternary complex to the ribosome (Steps 2, 4, 7, 10) and tRNA translocation (Steps 3, 6, 9, 12).

reduction) or $k_2$ (~fourfold reduction). The near identity of the red and black traces, performed at different eEF2·GTP concentrations, suggests that this factor interacts with both **1** and **2**, with a dissociation constant significantly less than 1 μM. The large inhibitory effect of eEF2·GTP on $k_{-1}$ is consistent with its role as a translocase, and with recent results demonstrating that a principal role of EF-G, the prokaryotic equivalent of eEF2, is to inhibit back-translocation (*Adio et al., 2015*). eEF2·GTP inhibition of $k_2$ may be due, at least in part, to a requirement for eEF2·GDP dissociation prior to Phe-TC binding.

Formation of Structure **4** from Structure **1**, as measured by the co-sedimentation assay, requires the presence of eEF2·GTP and proceeds at a considerably slower rate than formation of Structure **3** from Structure **1** (*Figure 3*), allowing estimation of a $t_{1/2}$ for Step 3, a second pseudo-translocation step involving conversion of **3** to **4,** of 210 ± 10 s. It is this further slow step that accounts for the lack of significant effect of preincubation with eEF2·GTP (5' or 60') on the rate of formation of **4** from **1** (*Figure 3*).

## Rates of oligopeptide formation and Met-tRNA[Met] cosedimentation

Using ribosomes programmed with the appropriate coding sequence mutants (*Supplementary file 1*) and [35S]-Met-TC, we employ a rapid mixing and quench assay to measure rates of PheMet, PheLysMet, and PheLysValMet synthesis, with detection and quantification of product by thin layer electrophoresis (TLE) (*Figure 4A* and *Figure 4—figure supplement 1*). For PheMet synthesis (*Figure 4B*) we preform Structure **4** and measure its conversion to Structure **6**. We measure PheLysMet synthesis, Structure **9**, starting from either Structure **4** or Structure **7** (*Figure 4C*) and

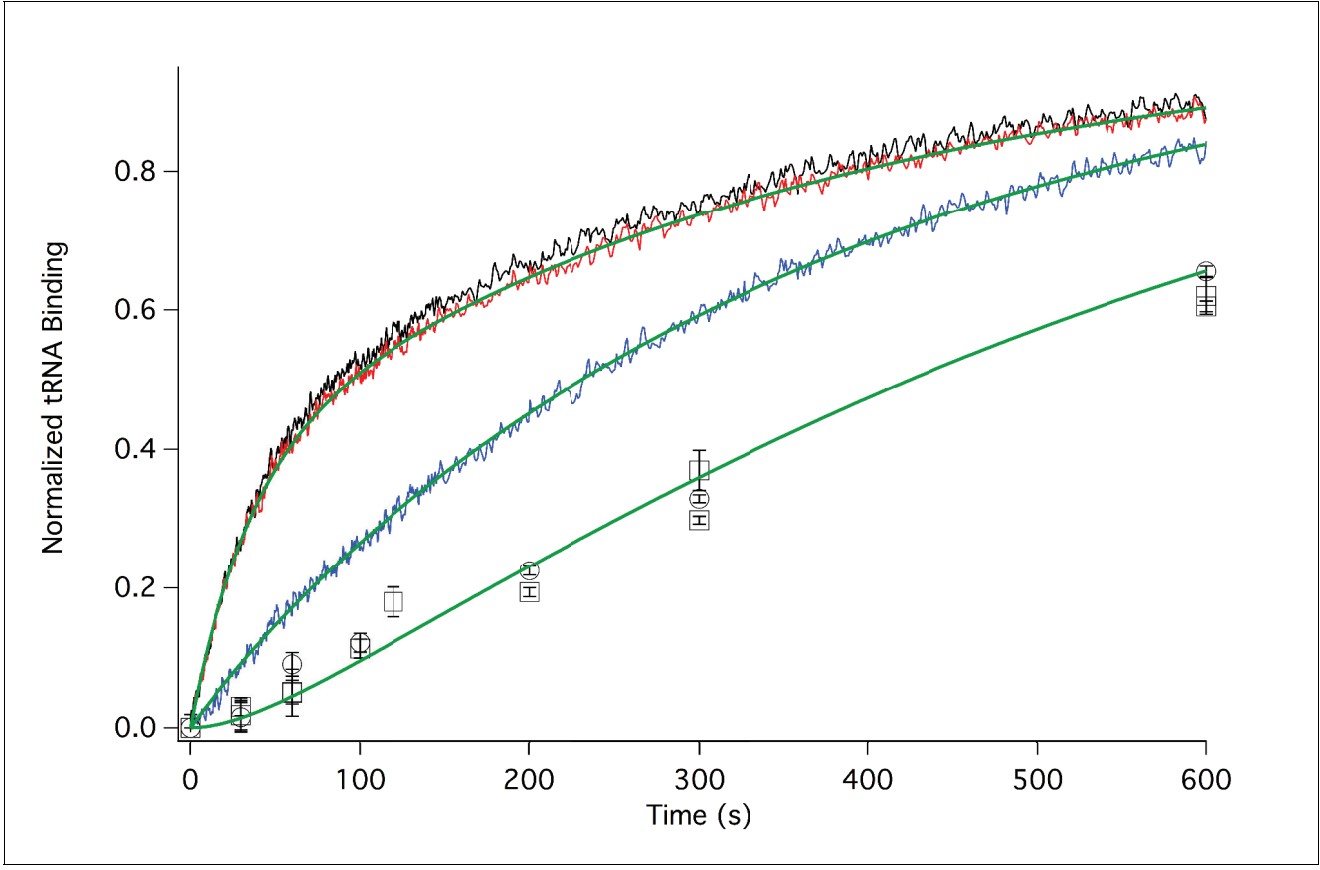

**Figure 3.** Rates of initial Phe-tRNA[Phe] binding measured by fluorescence anisotropy or Phe-tRNA[Phe] cosedimentation. Fluorescence anisotropy changes were monitored after rapid mixing of Phe-tRNA[Phe] (Prf) ternary complex (0.1 μM final concentration, containing 1 mM GTP) with 80S·FVKM-IRES complex (0.1 μM final concentration) either in the absence of eEF2 (blue line) or with 80S·FVKM-IRES complex that was pre-incubated with either 3 μM (black line) or 1 μM eEF2·GTP (red line) for 1–2 hr. These long times ensured full equilibration prior to TC addition. In the latter cases, eEF2 concentration was kept constant by including 3 μM or 1 μM eEF2, respectively, in the TC solution. eEF2 displays virtually no GTPase activity when it is not bound to the ribosome (*Nygård and Nilsson, 1989*). Rates of Phe-tRNA[Phe] binding to the P site, as determined by cosedimentation, were measured by rapidly mixing Phe-TC (1.6 μM final concentration) with 80S·FVKM-IRES complex (0.8 μM final concentration) pre-incubated for 5' – 60' in the presence (1 μM) (□) or absence (○) of eEF2·GTP. In both cases, eEF2 final concentration after mixing was adjusted to 1 μM, by including 1 μM or 2 μM eEF2·GTP, respectively, in the TC solution. After quenching with 0.5 M MES buffer (pH 6.0), ribosome bound Phe-tRNA[Phe] was measured by cosedimentation. In the preincubation experiment, three-fold increases of both eEF2·GTP and Phe-TC concentrations, or of just eEF2·GTP concentration, had little effect on the cosedimentation results. Results in this Figure are corrected for IRES-independent changes in fluorescence anisotropy or Phe-tRNA[Phe] cosedimentation (*Figure 3—figure supplements 1,2*). All three solid green lines are best fits of the results obtained to the scheme in *Figure 2*, using the numerical integration program Scientist.

The following figure supplements are available for figure 3:

**Figure supplement 1.** Corrected IRES-dependent time courses for initial Phe-tRNA[Phe] binding as measured by fluorescence anisotropy.

**Figure supplement 2.** Corrected IRES-dependent time courses for initial Phe-tRNA[Phe] binding as measured by Phe-tRNA[Phe] cosedimentation.

PheValLysMet synthesis, Structure **12**, starting from either Structure **7** or Structure **10** (*Figure 4D*). In all three cases, reactions involving only TC binding and a single peptide bond formation (**4** to **6**; **7** to **9**; **10** to **12**) proceed in remarkably similar fashion, each showing biphasic behavior with a rapid phase accounting for 65 ± 10% of reaction proceeding with a $t_{1/2}$ of ~6–9 s and a slower, minor phase proceeding much more slowly ($t_{1/2}$ ~220–240 s), possibly corresponding to defective ribosomes. Reactions involving formation of two peptide bonds, as in the conversion of **4** to **9** or **7** to **12** are well approximated as single phase reactions with $t_{1/2}$ values of 90–110 s. Conversion of **4** to **9** proceeds via Steps 4 – 8, allowing the $t_{1/2}$ value for the translocation Step 6 to be estimated as 84 s,

**Table 1.** Apparent rate constants for Steps 1 and 2.

| Apparent rate constants ($s^{-1}$) | -eEF2 | +eEF2 |
| --- | --- | --- |
| $k_1$ | 0.0071 ± 0.0033 | 0.0033 ± 0.0001 |
| $k_{-1}$ | 0.15 ± 0.04 | 0.0034 ± 0.0001 |
| $k_2$ ([Phe-TC] = 0.1 µM) | 0.11 ± 0.04 | 0.0256 ± 0.0002 |

from the difference between the $t_{1/2}$ value for the **4** to **9** reaction and the sum of the $t_{1/2}$ values for the **4** to **6** and **7** to **9** reactions (major phases). Similarly, the $t_{1/2}$ value for the translocation Step 9 can be estimated as 110 s from the difference between the $t_{1/2}$ value for the **7** to **12** reaction and the sum of the $t_{1/2}$ values for the **7** to **9** and **10** to **12** reactions. Since the di-, tri- and tetrapeptides synthesized in the results reported in *Figure 4* use different coding sequence mutants, these estimates of translocation $t_{1/2}$ values depend on the not unreasonable assumption that the identities of the tRNAs undergoing translocation do not have a major influence on the translocation rate. With this caveat, the results presented in *Figure 4* lead to the clear conclusion that translocation is the rate limiting step in each of the first two cycles of polypeptide elongation, proceeding from **4** to **10**.

In an attempt to resolve the TC binding step (reactions 4, 7, and 10) from the peptide formation step (reactions 5, 8, and 11) we also employed a rapid mixing and quench assay to determine the rates with which [$^{35}$S]-Met-tRNA$^{Met}$ is able to cosediment with the ribosome following mixing of [$^{35}$S]-Met-TC with structures **4, 7,** or **10**. This strategy was successful for [$^{35}$S]-Met–TC reaction with structure **7** (containing P-site bound PheLys-tRNA$^{Lys}$, *Figure 4C*) or structure **10** (containing P-site bound PheValLys-tRNA$^{Lys}$ *Figure 4D*), in which the [$^{35}$S]-Met-TC cosedimentation rates outpace the rates of peptide bond formation with Met-TC. These rate differentials permit estimates to be made for the $t_{1/2}$ values of TC binding (Step 7, 3 s; Step 10, 2 s) and peptide bond formation (Step 8, 4 s; Step 11, 7 s). They also provide a clear indication that, within Structures **8, 9, 11** and **12**, Met-tRNA-$^{Met}$, PheLysMet-tRNA$^{Met}$, and PheValLysMet- tRNA$^{Met}$, whenbound to the A-site, efficiently cosediment with ribosomes, which is typical for A-site bound tRNAs in conventional (non-IRES) elongation complexes (*Warner and Rich, 1964*; *Nwagwu, 1975*).

However, for [$^{35}$S]-Met–TC reaction with structure **4** (containing P-site bound Phe-tRNA$^{Phe}$), the [$^{35}$S]-Met-TC cosedimentation rate is much slower than the dipeptide formation rate (*Figure 4B*). This indicates that PheMet-tRNA$^{Met}$, and possibly Met-tRNA$^{Met}$ as well, are not bound stably to the ribosome in Structures **5** and **6,** and that only PheMet-tRNA$^{Met}$ bound to the P-site (Structure **7**) is fully recovered by cosedimentation. As a result, the cosedimentation assay does not permit estimation of the $t_{1/2}$ values for Steps 4 and 5. It is possible that the lability of the A-site tRNAs in structures **5** and **6** is due to IRES binding to the E-site, which is absent in structures **8, 9** and **11, 12,** and may reflect an allosteric A-site: E-site interaction. Evidence for allosteric A-site/E-site interactions has been presented for both bacterial and eukaryotic ribosomes (*Nierhaus 1990*; *Chen et al., 2011*; *Ferguson et al., 2015*), although the general validity of this interaction has been questioned (*Semenkov et al., 1996*; *Petropoulos and Green, 2012*).

## Translocation of tetrapeptidyl-tRNA (Step 12) is much more rapid than of tripeptidyl-tRNA (Step 9)

The results presented in *Figure 4* show that translocation proceeds slowly through the first two elongation cycles of nascent protein synthesis, raising the question of how far nascent protein synthesis has to proceed to overcome the retarding effect of ribosome-bound IRES. In *Figure 5* we present the results of two experimental approaches demonstrating that translocation of tetrapeptidyl-tRNA proceeds much more rapidly than translocation of tripeptidyl-tRNA.

The first approach makes use of the fact that formation of peptidyl-puromycin proceeds more rapidly with peptidyl-tRNA bound to the P-site than to the A-site, permitting puromycin reactivity to distinguish A-site from P-site peptidyl-tRNA. As shown in *Figure 5A*, puromycin (1 mM) reacts with A-site bound PheValLys-tRNA$^{Lys}$, Structure **9**, about 20times more slowly ($t_{1/2}$ 1400 ± 300 s) than it reacts with P-site bound PheValLys-tRNA$^{Lys}$($t_{1/2}$ 76 ± 16 s). The corresponding $t_{1/2}$ value for puromycin reaction with PheValLys-tRNA$^{Lys}$ undergoing translocation from the A- to P-site is 170 ± 30 s.

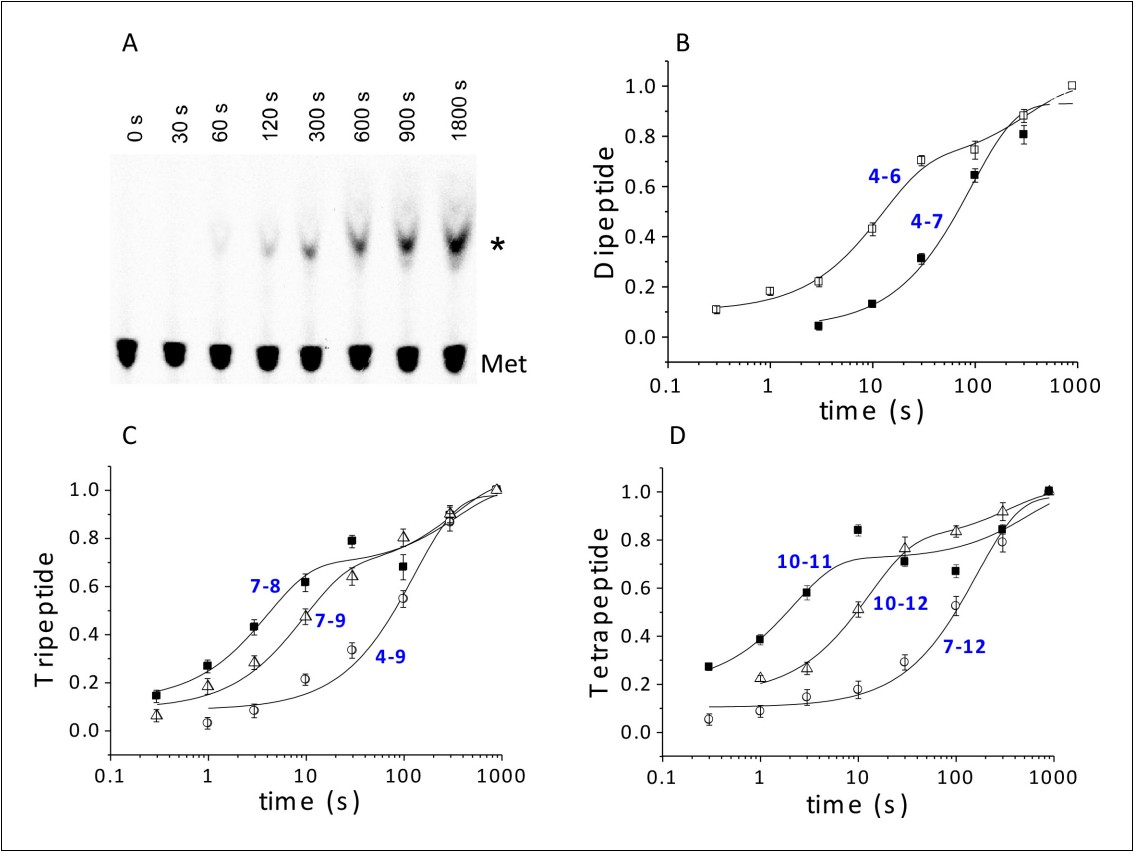

**Figure 4.** Kinetics of peptide synthesis and Met-tRNA$^{Met}$ cosedimenting with ribosomes. Reaction mixtures were quenched at various times after mixing. Peptide synthesis aliquots were quenched with 0.8 M KOH, and the released [$^{35}$S]-containing peptide was resolved and quantified by TLE and autoradiography (Materials and methods). Cosedimentation assay aliquots were quenched with with 0.5 M MES buffer (pH 6.0) and [$^{35}$S] cosedimenting with ribosomes was determined. For all the reactions shown, final concentrations of reactants after mixing were: 80S·IRES complexes (0.8 µM); all added TCs (1.6 µM); eEF2·GTP (1 µM). The numbers in blue in parts (B–D) refer to the Structures in *Figure 2* whose rates of conversion are measured. For example, the peptide synthesis result in part (B) labeled 4 – 6 measures conversion of Structure 4 to Structure 6. (A) Time course for formation of PheValLysMet tetrapeptide as determined by TLE. 80S·FVKM-IRES complex was mixed with Phe-TC, Val-TC, Lys-TC and [$^{35}$S]-Met-TC. The migration positions of [$^{35}$S]-Met and [$^{35}$S]-PheValLysMet (*) are indicated. (B) 80S·FM-IRES complexes with Phe-tRNA$^{Phe}$ at the P site were mixed with [$^{35}$S]-Met-TC. Dipeptide synthesis (□); cosedimentation assay (■). (C) Tripeptide synthesis: 80S·FKM-IRES complexes with either Phe-tRNA$^{Phe}$ (O) in the P site (Structure 4) or PheLys-tRNA$^{Lys}$ (Δ) in the P site (Structure 7) were mixed with either Lys-TC and [$^{35}$S]-Met-TC or with just [$^{35}$S]-Met-TC, respectively. Cosedimentation assay: 80S·FKM-IRES complex with PheLys-tRNA$^{Lys}$ in the P site was mixed with [$^{35}$S]-Met-TC (■). (D) Tetrapeptide synthesis: 80S·FVKM-IRES complexes with either PheVal-tRNA$^{Val}$ (O) in the P site (Structure 7) or PheValLys-tRNA$^{Lys}$ (Δ) in the P site (Structure 10) were mixed with either Lys-TC and [$^{35}$S]-Met-TC or with just [$^{35}$S]-Met-TC, respectively. Cosedimentation assay: 80S·FKM-IRES complex with PheValLys-tRNA$^{Lys}$ in the P site was mixed with [$^{35}$S]-Met-TC (■). Solid lines are best fits using single (*B*, 4–7; *C*, 4–9; *D*, 7–12) or double (*B*, 4–6; *C*, 7–8, 7–9; *D*, 10–11, 10–12) exponentials.

The following figure supplements are available for figure 4:

**Figure supplement 1.** Time courses for formation of PheMet dipeptide and PheLysMet tripeptide as determined by TLE.

**Figure supplement 2.** Added 30S carrier does not significantly change the amount of FVKM-tRNA$^{Met}$ co-sedimenting with 80S ribosomes in the presence and absence of FVKM-IRES.

This increase of approximately 100 s for translocating PheValLys-tRNA$^{Lys}$ vs. translocated PheValLys-tRNA$^{Lys}$ closely matches the $t_{1/2}$ value of 110 ± 30 s estimated above for the translocation of tripeptidyl-tRNA (*Table 2*) and can be assigned to the translocating step. In contrast, the rates of puromycin reaction with translocating and translocated PheValLysMet-tRNA$^{Met}$ (Structure 13) are indistinguishable from one another ($t_{1/2}$ values of 37 ± 4 s and 46 ± 7 s, respectively, *Figure 5B*), a clear demonstration that translocation of PheValLysMet-tRNA$^{Met}$ proceeds rapidly with respect to

**Table 2.** $t_{1/2}$ values*.

| Step (s) | $t_{1/2}$ (s) |
|---|---|
| 1[†] | 230 ± 5 |
| 1 (+eEF2)[†] | 237 ± 5 |
| 2[‡] | 15 ± 9 |
| 2 (+eEF2)[‡] | 30 ± 5 |
| 3 | 210 ± 10 |
| 4 + 5 | 8 ± 2 |
| 4-8 | 98 ± 15 |
| 6 = (4-8) − (4+5) − (7+8)[§] | 84 ± 16 |
| 7 | 3 ± 1 |
| 8 = (7+8) − 7[§] | 4 ± 2 |
| 7 + 8 | 6 ± 2 |
| 7-11 | 128 ± 26 |
| 9 = (7-11) − (7+8) − (10+11)[§] | 110 ± 30 |
| 10 | 2 ± 1 |
| 11 = (10 + 11) − 10[§] | 7 ± 3 |
| 10 + 11 | 9 ± 2 |
| 12 | <10 |

\* Error ranges shown are based on the variances of fits to single or double exponentials of the results presented in *Figure 4*, unless otherwise noted.

[†] Calculated as 0.69 $(k_{-1} + k_2)/k_1 k_2$ (see **Table 1**).

[‡] Calculated as 0.69 $(k_{-1} + k_2)/k_2^2$ (see **Table 1**).

[§] Error ranges for these steps, which are not observed directly, are based on the error ranges of the directly observed steps.

puromycin reaction. Our results allow us to estimate an upper limit value of $t_{1/2}$ for the translocation Step 12 of ≤10 s.

Puromycin reacts at similar rates with translocated PheValLys-tRNA[Lys] (Structure **10**, $t_{1/2}$ 76 ± 16 s) and PheValLysMet-tRNA[Met] (Structure **13**, $t_{1/2}$ 46 ± 7 s). These rates, while consistent with those reported by others for puromycin reaction with eukaryotic P-site bound Met-tRNA[Met] (*Lorsch and Herschlag, 1999*), N-AcPhe-tRNA[Phe] (*Ioannou et al., 1997*), and Cy3-Met-tRNA[Met] (*Ferguson et al., 2015*), are several hundred-fold slower than those measured for puromycin reaction with prokaryotic P-site bound peptidyl- or fMet-tRNA. This largely explains why the rate reduction for puromycin reaction with A-site *vs.* P-site bound peptidyl-tRNA is so much more modest for eukaryotic ribosomes (~20-fold, *Figure 5A*) than for prokaryotic ribosomes ($10^3$–$10^4$-fold, *Pan et al., 2007*; *Semenkov et al., 1992* ; *Sharma et al., 2004*; *Peske et al., 2004* ).

Above we have demonstrated that, under our conditions, aa-tRNA binding and peptide bond formation proceed with an overall $t_{1/2}$ of 6 – 9 s for each of the three elongation steps we have studied. This relative constancy, coupled with the much slower translocation of tripeptidyl-tRNA (Step 9) *vs.* tetrapeptidyl-tRNA (Step 12), leads to the prediction that synthesis of a longer peptide that required the tripeptidyl-tRNA translocation step (Step 9) would proceed significantly more slowly than synthesis not requiring this step.

In the second approach we verified this prediction by demonstrating that octapeptide FKVRQWLM formation, as measured by the cosedimentation assay, is much slower when synthesis is initiated with P-site bound PheLys-tRNA[Lys] (Structure **7**) *vs.* P-site bound PheLysVal-tRNA[Val] (Structure **10**) (*Figure 5C*). Indeed, the rates of FKVRQWLM synthesis are only marginally increased when reaction is initiated with P-site bound tetrapeptidyl-tRNA or pentapeptidyl-tRNA as compared with tripeptidyl-tRNA, reinforcing the notion that the retarding effect of ribosome-bound IRES on protein synthesis is largely overcome following translocation of tripeptidyl-tRNA.

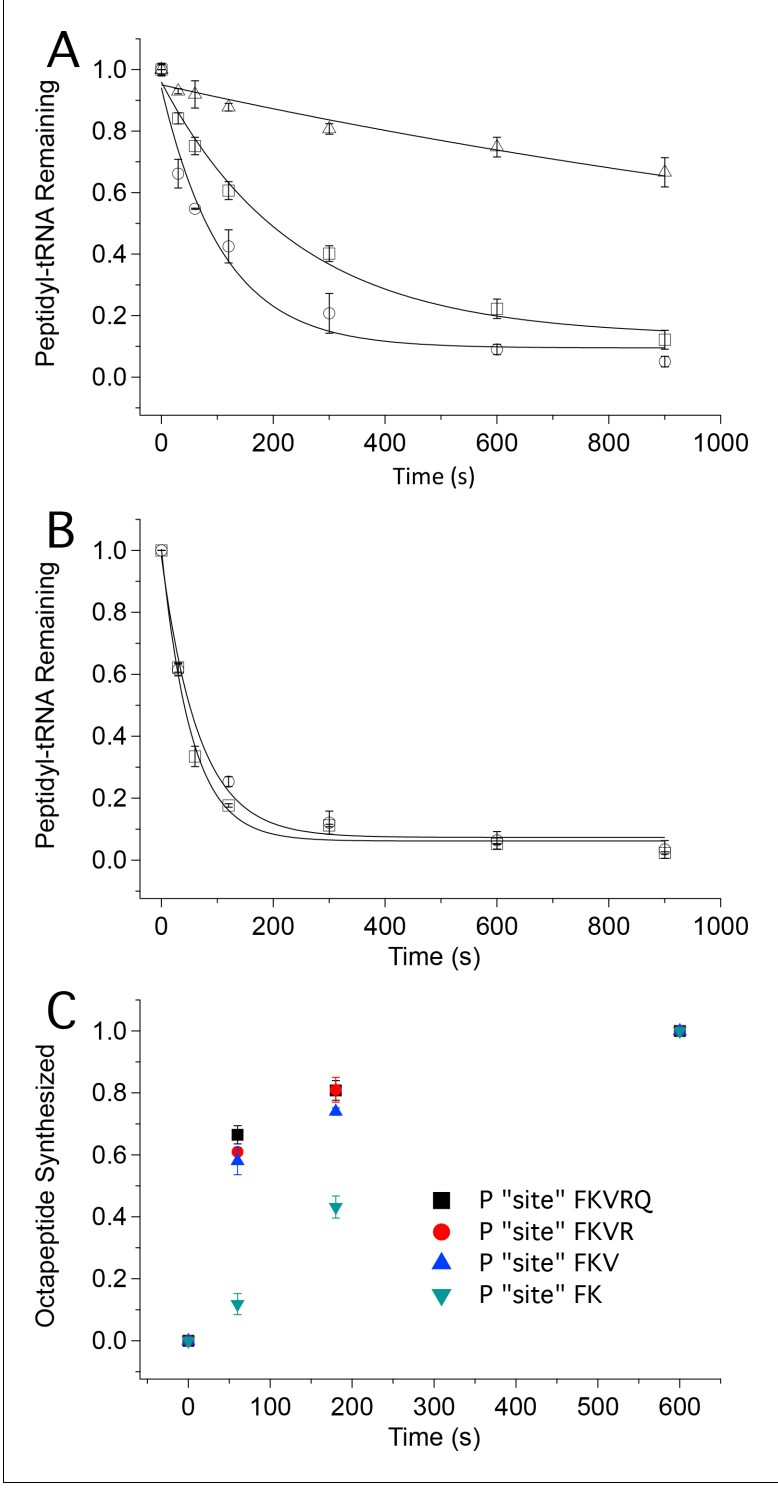

**Figure 5.** Tetrapeptide translocation (Step 12) is faster than tripeptide translocation (Step 9). (**A**) Puromycin reaction with PheValLys-tRNA$^{Lys}$ bound either at the A site (**D**) or at the P-site (O) of the 80S·FVKM-IRES complex or being translocated from the A site to the P site (□). (**B**) Puromycin reaction with PheValLysMet-tRNA$^{Met}$ either bound at the P-site (O) of the 80S·FVKM-IRES complex or being translocated from the A site to the P site (□). Lines in **A**. and **B**. Are fits to single exponentials. (**C**) Time dependence of PheLysValArgGlnTrpLeuMet octapeptide synthesis from the 80S·FKVRQWLM-IRES complex containing various peptidyl-tRNAs pre-bound at the P site, as indicated. The pre-bound peptidyl tRNAs were prepared using the standard procedure (see Complex Preparations in Materials and methods) by incubating the 80S-IRES complex with the relevant TCs for 15 min. The

*Figure 5 continued on next page*

*Figure 5 continued*

remaining TCs needed for octapeptide synthesis, including [$^{35}$S]-Met-TC, were then added, each at a concentration of 1.6 μM, for the indicated times prior to quenching. PheLysValArgGlnTrpLeuMet octapeptide synthesis was measured by [$^{35}$S]-Met cosedimenting with 80S ribosome.

The following figure supplement is available for figure 5:

**Figure supplement 1.** Octapeptide synthesis: 80S·FKVRQWLM-IRES complex with FKVRQWLM-tRNA$^{Met}$ in the P-site was prepared using the standard procedure (see Complex Preparations in Materials and methods) and incubating the 80S-IRES complex with the eight relevant TCs (including [$^{35}$S]-Met-TC) for 40 min.

## Discussion

The results presented in this paper constitute the first time that rates of reaction have been determined for the initial cycles of IRES-dependent elongation. They demonstrate quite clearly that the first two cycles of elongation proceed much more slowly than subsequent steps, and that these reduced rates arise from slow, rate-determining, pseudo-translocation and translocation steps. Translocation during the first elongation cycle (Step 6) clearly requires displacement of the IRES from the E-site, so it is not unexpected that it would be slow. Less predictable is the slow translocation in the second elongation cycle, (Step 9) after the IRES structure has, presumably, already left the E-site (*Figure 1*). The slow rate of Step 9 might be due to a full dissociation of IRES from the ribosome during this step, a suggestion that could be tested by appropriately designed structural studies. In any case, our results do clearly demonstrate that, following translocation of tripeptidyl-tRNA from the A- to P-site, the pace of nascent peptide chain elongation picks up dramatically. Further work, comparing quantitatively the rates of successive cycles of nascent peptide elongation following tetrapeptide formation (i.e, cycles 4, 5, 6, 7, etc.) will be required to determine how many cycles are required before any retarding influence of bound CrPV-IRES is completely eliminated.

Our results also clarify an aspect of the initial binding of the first aa-tRNA to the 80S·CrPV-IRES complex. Prior results have shown that initial aa-tRNA binding, in the form of a ternary complex, to an 80S·IRES complex, as measured either by cosedimentation (*Fernandez et al., 2014*), or by filter binding and toeprinting (*Yamamoto et al., 2007*), requires eEF2·GTP, leading to the conclusion that initial aa-tRNA binding can only bind to the 80S·IRES complex after an eEF2-dependent translocation event (*Fernandez et al., 2014*). While we agree with the experimental results, and have in fact reproduced the cosedimentation result in our own work, we disagree with the conclusion. This is because these earlier experiments only measured stable aa-tRNA binding, corresponding to formation of Structure **4** in which aa-tRNA binds to the P-site. However, it is clear from the anisotropy experiment conducted in the absence of added eEF2·GTP (*Figure 3*, blue trace) that ternary complex binding measured in situ, which can monitor labile binding to the A-site (Structure **3**) does not require eEF2·GTP. This is easily understood as an example of Le Chaltelier's principle, in which the equilibrium between Structure **1** (closed A- site) and Structure **2** (open A- site), which strongly favors Structure **1**, is pulled to the right by aa-tRNA binding. Preincubation with eEF2 also shifts the equilibrium to the right, leading to an initial rapid phase of reaction with Phe-TC (*Figure 3*).

This latter shift, for which the results presented in *Figure 3* provide strong inferential evidence, appears to be at odds with earlier toeprinting results showing no shift in IRES position within the 80S·CrPV-IRES complex on addition of eEF2 alone (*Pestova et al., 2003*; *Jan et al., 2003*). In agreement with the suggestion of *Muhs et al. (2015)*, we believe it likely that this apparent inconsistency arises from eEF2 dissociation from the ribosome during the toeprinting assay (*Pestova et al., 1996*), with the consequent favoring of Structure **1**. This is because GTP is required for tight binding of eEF2 to the ribosome (*Nygård and Nilsson, 1984*), but the toeprinting assay is carried out for an extended period of time (45 min) under non-denaturing conditions in the absence of added GTP, conditions that would eventually deplete GTP due to ribosome-dependent eEF2·GTP hydrolysis (*Nygård and Nilsson, 1989*). In addition, the toeprinting assay is performed at a Mg$^{2+}$ concentration of 10.5 mM, considerably higher than the 5 mM used in our kinetic studies, which could also affect the **1** to **2** equilibrium position.

How relevant are the present results for in vivo initiation of IRES-dependent protein synthesis? We note three potential concerns. First, our in vitro system is quite heterogeneous, with ribosomes

derived from shrimp cysts, yeast elongation factors, and yeast and *E. coli* charged tRNAs. However, as reviewed in *Koh et al. (2014)*, IRESs can initiate translation on ribosomes from many eukaryotic organisms, including shrimp (*Cevallos and Sarnow, 2005*), indicating that the molecular mechanism is not species-specific. CrPV IRESs in particular can initiate translation on ribosomes from yeast (*Thompson et al., 2001*) to human (*Spahn et al., 2004*). Furthermore, eukaryotic elongation factors have structures that are very strongly conserved (*Soares et al., 2009*; *Jørgensen et al., 2006*), and there is strong evidence that charged tRNAs from one species form functional complexes with both eEF1A and ribosomes from a different species (*Jackson et al., 2001*; *Ferguson et al., 2015*). Second, the coding sequences employed in this work are different from that immediately downstream of wt-CrPV-IRES (*Supplementary file 1*). This is also unlikely to pose a major difficulty, given the strong evidence that mutations in the downstream sequence are, in general, tolerated without substantial effect on initiation of translation (*Tsukiyama-Kohara et al., 1992*; *Wang et al., 1993*; *Hellen and Sarnow, 2001*; *Rijnbrand et al., 2001*), although mutations of some downstream sequences do give rise to relatively minor changes in IRES activity (*Kim et al., 2003*; *Shibuya et al., 2003*; *Wang et al., 2013*). Third, the elongation rate of even the later cycles of IRES-dependent elongation (*Figure 5C*) is quite slow (~0.1 s$^{-1}$). Although this rate is essentially identical to that reported for tripeptide synthesis in a cap-dependent yeast-based in vitro translation system which requires five initiation factors and eEF3 in addition to eEF1A and eEF2 (*Acker et al., 2007*; *Eyler and Green, 2011*; *Gutierrez et al., 2013*), it is 1.5–2 orders of magnitude slowerthan rates of peptide elongation that have been estimated for intact eukaryotic cells at 37°C (3–10 s$^{-1}$) (*Boehlke and Friesen, 1975*; *Hershey, 1991*). There is evidence that, in many eukaryotic cells, the protein synthesis machinery is highly organized, containing several components, including ribosomes, a multi-aminoacyl-tRNA synthetase complex, eEF-1A, and several auxiliary proteins (*Negrutskii et al., 1994*; *Negrutskii and El'skaya, 1998*; *David et al., 2011*). It has been suggested that this organized structure optimizes translation rate by coordinating synthetase activities to facilitate channeling of aa-tRNAs to the elongating ribosomes. Thus, protein synthesis in a permeabilized mammalian cell, in which this structure is likely to be preserved, proceeds 40-fold faster than what is obtained in a cell-free system prepared from the same cells which presumably lacks this structure (*Negrutskii et al., 1994*). The slow rates measured for both the IRES-dependent and cap-dependent in vitro systems could be due, at least in part, to their lack of aa-tRNA channeling. Such channeling would be unlikely to accelerate the very slow translocation rates in the initial peptide elongation cycles reported in this work, although we cannot exclude the possibility that other proteins present in vivo might have such effects. Future efforts will address this issue. Here, incorporation of some of the features of a recently introduced in vitro protein synthesissystem in which initiation is carried out using the IRES from hepatitis C virus could be useful (*Machida et al., 2014*).

Detailed mechanistic characterization of many aspects of eukaryotic polypeptide elongation has been held back by the lack of a convenient system for its study. The very simple in vitro IRES-dependent elongation system described here should be useful in overcoming this limitation. As one example, it is generally assumed, based on extensive structural similarities (*Jørgensen et al., 2006*), that eEF2 functions in catalyzing eukaryotic elongation in much the same way that EF-G catalyzes prokaryotic elongation, but this assumption does not take into account some important structural differences, including the fact that eEF2 is subject to post-translational modifications not found in EF-G, with clear consequences for activity but, as yet, little understanding of mechanism (*Dever and Green, 2012*; *Mittal et al., 2013* ; *Greganova et al., 2011*; *Liu et al., 2012*). The CrPV-IRES based system should permit detailed rate and structural dynamic studies of eEF2 catalytic function, of the kind that have proved so useful in elucidating EF-G function in bacterial protein synthesis (*Pan et al., 2007*; *Chen et al., 2011*; *Chen et al., 2013*; *Holtkamp et al., 2014*; *Salsi et al., 2015*).

## Materials and methods

### Plasmid construction and cloning

The wt CrPV Phe-IRES vector, as well as several variants in which the first Ala codon is replaced by a Phe codon, were the kind gifts of Dr. Eric Jan.This replacement, which has little effect on the initiation of translation (see Discussion and *Figure 1—figure supplement 1*), was made as a matter of convenience, since tRNA$^{Phe}$ was available to us and the appropriate tRNA$^{Ala}$ acceptor was not. The

vectors encoding the PheMet, PheValMet, PheValLysMet, and PheLysValArgGlnTrpLeuMet were generated by PCR insertion of corresponding sequences (*Supplementary file 1*) into the CrPV Phe-IRES vector. All cloned sequences were verified by standard sequencing methods using appropriate primers.

## In vitro transcription

For in vitro transcription of full-length mRNA for the Luciferase assay, the WT and mutated Phe-IRES plasmids were linearized with *XbaI*, which cleaves the plasmids after the firefly luciferase coding region. mRNA was transcribed in vitro using the AmpliScribe T7 transcription kit (EPICENTRE) according to the manufacturer. For in vitro transcription of short-length mRNAs, the mutated IRES plasmids were linearized with *NarI*, which cleaves 33 nt downstream of the ATG start codon of the luciferase coding region.

## Luciferase assay

In vitro translation of firefly luciferase with WT and mutated F-IRES mRNA (1 µg in 50 µL of reaction mixture) was performed using the Flexi Rabbit Reticulocyte Lysate System (Promega) according to the manufacturer. IRES mRNA was omitted in the control reaction. Fluc activities (*Figure 1—figure supplement 1*) were determined using a plate reader (Envision 2103, Perkin-Elmer) to detect the luminescence signal.

## Ribosomes, elongation factors and tRNAs

Shrimp (*A. salina*) 80S ribosomes were prepared from dried, frozen cysts as previously described (*Iwasaki and Kaziro, 1979*) with some modifications. After the shrimp cysts were ground open, debris was removed by centrifugation at 30,000xg for 15 min and crude 80S ribosomes were precipitated from the supernatant by addition of 4.5% (w/v) PEG 20K (*Ben-Shem et al., 2011*). 40S and 60S subunits were resolved on 10–30% sucrose gradients after puromycin treatment. *E. coli* 30S subunits were prepared as described (*Grigoriadou et al., 2007*). eEF1A was purified from yeast according to published methods (*Thiele et al., 1985*). His6-eEF2 was isolated from an overexpressing yeast strain (TKY675) generously provided by Dr. Terri Kinzy, and purified as described (*Jørgensen et al., 2002*). Proflavin-labeled Phe-tRNA$^{Phe}$, denoted Phe-tRNA$^{Phe}$(prf), was prepared as previously described (*Wintermeyer and Zachau 1974*, *Betteridge et al., 2007*). Yeast tRNA$^{Phe}$ was purchased from Sigma. Other isoacceptor tRNAs were prepared from bulk tRNA (Roche) from either *E. coli* (tRNA$^{Gln}$, tRNA$^{Lys}$, tRNA$^{Met}$) or yeast (tRNA$^{Arg}$, tRNA$^{Leu}$, tRNA$^{Trp}$, tRNA$^{Val}$) by hybridization to immobilized complementary oligoDNAs, as described (*Barhoom et al., 2013* ; *Liu et al., 2014*). *E. coli* and yeast tRNAs were charged with their cognate amino acids as described (*Pan et al., 2006*, *2009*).

## Complex preparations. TCs and various 80S·IRES complexes

All complexes were prepared in buffer 4 (40 mM Tris-HCl pH 7.5, 80 mM NH$_4$Cl, 5 mM MgOAc$_2$, 100 mM KOAc, 3 mM 2-mercaptoethanol) at 37°C. For the preparation of ternary complexes (TC, aa-tRNA·eEF1A·GTP) and 80S·IRES complexes containing either Phe-tRNA$^{Phe}$ or peptidyl-tRNA bound in the P-site, buffer 4 was supplemented with 1 mM GTP and 1 mM ATP. All TC complexes were prepared by incubating the relevant charged tRNA (1.6 µM, based on amino acid stoichiometry) with eEF1A (8 µM) for 5 min. 80S·IRES complexes were formed by incubation of shrimp 40S (0.8 µM) and 60S (1.6 µM) subunits with the appropriate IRES (2.4 µM) for 5 min. 80S·IRES complexes containing Phe-tRNA$^{Phe}$ or peptidyl-tRNA bound in the P-site were formed by mixing 80S·IRES complexes (0.8 µM) with 1 µM eEF2 and the appropriate TCs (1.6 µM for each) for 15–40 min. To determine radioactively labeled aa-tRNA binding stoichiometries, 40 µL samples were subjected to ultracentrifugation at 4°C (540,000xg) for 40 min through a 1.1 M sucrose cushion. Excess bacterial 30S bacterial ribosome subunits (600 pmol/15 ± 5 µL) were added as carrier to enhance pelleting and allow facile calculation of complex recovery. Control experiments carried out in the absence of IRES or of both IRES and 80S ribosomes demonstrated that only negligible amounts of labeled peptidyl-tRNA cosedimented due to binding to 30S subunits (*Figure 4—figure supplement 2*). The pellets were gently washed twice with buffer 4 and dissolved in 100 µL of buffer 4 for A$_{260}$ determination. Ribosome recoveries typically varied between 60 and 80%.

## Kinetic measurements

Unless otherwise noted, all reactions were performed at 37°C in buffer 4 supplemented with 1 mM GTP. All kinetic results reported are the averages of 2–4 independent determinations, performed on different days. No systematic effort was made to carry out duplicate experiments using independently made stock reagent solutions, although this was sometimes done. Error bars in figures are shown as average deviations.

## Rates of Phe-TC binding by fluorescence anisotropy change (*Figure 3*)

Phe-tRNA$^{Phe}$(prf)·eEF1A·GTP ternary complex was rapidly mixed with 80S·FVKM-IRES complex in the presence or absence of eEF2·GTP using a KinTek stopped-flow spectrofluorometer model SF-300X. Proflavin labeled Phe-tRNA$^{Phe}$ was excited at 462 nm and monitored using a pair of 495 nm long-pass filters. A T-shape configuration was utilized such that instrument-specific polarizers were attached to both the excitation and the two emission light paths. In each independent measurement, 15–20 shots (rapid mixing of samples) were averaged to provide the time course of anisotropy change. The g-factor and anisotropy value were calculated using the instrument software as described (*Lakowicz 1999*, *Ameloot et al., 2013*). Experimental data were processed and analyzed by Felix software (from PTI).

## Rates of [$^3$H]-Phe-TC or [$^{35}$S]-Met-TC binding by cosedimentation

80S·IRES complex (0.8 µM) with no tRNA bound (*Figure 2*) was rapidly mixed with [$^3$H]-Phe-TC in the presence of eEF2·GTP in a KinTek Corporation RQF-3 Rapid Quench-Flow Instrument the reaction mixture was quenched at various times with 0.5 M MES buffer (pH 6.0), and the stoichiometry of ribosome-bound [$^3$H]-Phe-TC was determined by ultracentrifugation as described above for complex characterization. Similar procedures were used to determine the kinetics of [$^{35}$S]-Met-TC binding to 80S·IRES complexes containing Phe-tRNA$^{Phe}$ or peptidyl-tRNA in the P-site (*Figure 4*).

## Rates of peptide synthesis

### Di-, Tri- and Tetrapeptide

80S·IRES complexes containing either Phe-tRNA$^{Phe}$ or the appropriate peptidyl-tRNA in the P-site, prepared using the standard procedure (see Complex preparations above), were rapidly mixed with [$^{35}$S]-Met-TC (1.6 µM) and additional TCs as required (all 1.6 µM) in a KinTek Corporation RQF-3 Rapid Quench-Flow Instrument, and the reaction mixture was quenched at various times with 0.8 M KOH. [$^{35}$S]-Met-containing peptide was released from tRNA$^{Met}$ by further incubation at 37°C for 3 hr. The pH of the samples were adjusted with acetic acid to pH 2.8, lyophilized, suspended in water, and centrifuged to remove particulates, which contained no $^{35}$S. The supernatant was analyzed by thin layer electrophoresis as previously described (*Youngman et al., 2004*), using the same running buffer, and the labeled peptide was located by autoradiography. The identities of PheMet, PheLysMet, and PheValLysMet, (*Figures 4A*, *Figure 4-figure supplement 1*, *Figure 5C*) were confirmed by their comigrations with authentic samples obtained from GenScript (Piscataway, NJ). A further demonstration of tetrapeptide identity was provided by matrix-assisted laser desorption/ionization (MALDI) mass spectrometric analysis (Ultraflex III TOF/TOF, Bruker: Phe-Val-Lys-Met(Na+), calculated, 546.7; found 546.6. In addition, the 80S·IRES complex containing P-site bound Phe-Val-Lys-Met-tRNA was reacted for 40 min with 10 mM puromycin (37°C, buffer 4 plus 1 mM GTP). The resulting puromycin adduct, Phe-Val-Lys-Met-puro(H+), released into solution, was also identified by MALDI: calculated, 978.7; found, 978.9.

### Octapeptide

80S·FKVRQWLM-IRES complexes containing the appropriate peptidyl-tRNA in the P-site, prepared using the standard procedure (see Complex preparations above)were mixed with [$^{35}$S]-Met-TC (1.6 µM) and additional TCs as required (all 1.6 µM), for various times followed by quenching with 0.5 M MES (pH 6.0) buffer. PheLysValArgGlnTrpLeuMet octapeptide synthesis was measured by [$^{35}$S]-Met cosedimentation with 80S·FKVRQWLM-IRES complexes. Over the time scale of these measurements (60 –600 s, *Figure 5C*), all [$^{35}$S]-Met-tRNA$^{Met}$ stably bound to the ribosome undergoes a peptide transfer reaction (see *Table 2*). PheLysValArgGlnTrpLeuMet peptide was released from tRNA$^{Met}$ using the base treatment described above for other peptidyl tRNAs and its identity was confirmed

by its comigration during thin layer electrophoresis with an authentic sample obtained from Gen-Script (Piscataway, NJ) (*Figure 5C—figure supplement 1*).

### Rates of puromycin adduct formation

Rates of puromycin adduct formation were measured for FVK-tRNA$^{Lys}$ bound in the A-site, and for both FVK-tRNA$^{Lys}$ and FVKM-tRNA$^{Met}$ either pre-bound in the P-site or undergoing translocation from the A-site to the P-site. In all cases, reaction mixtures were quenched with 0.5 M MES (pH 6.0) buffer. The quenched samples were next ultracentrifuged and the radioactivity co-sedimenting with the ribosome, which decreases as more puromycin adduct is formed, was determined. All incubations and reactions were carried out at 37°C. All complexes were reacted with puromycin (1 mM, final concentration) for various times before quenching. No decreases in radioactivity co-sedimenting with the ribosome were observed in the absence of added puromycin.

### FVK-tRNA$^{Lys}$ bound in the A-site (Structure 9, PRE-3)

80S·FVKM-IRES complex with FVK-tRNA pre-bound at A-site was formed by incubating the 80S·FVKM-IRES complex containing FV-tRNA at the P-site (Structure **7**, POST-2, 0.4 μM), purified by sedimentation through a sucrose cushion, with 0.8 μM [$^3$H]-Lys-TC for one minute. The resulting complex was then reacted with puromycin.

### FVK-tRNA$^{Lys}$ bound in the P-site (Structure 10, POST-3)

Two procedures were employed, which yielded equivalent results. Procedure 1: 80S·FVKM-IRES complex containing FV-tRNA at the P-site (Structure **7**, POST-2, 0.4 μM), purified as described above, was incubated with 0.8 μM [$^3$H]-Lys-TC and eEF2·GTP (1.0 μM) for 15 min at 37°C. The resulting complex was then reacted with puromycin. Procedure 2: 80S·FVKM-IRES complex (0.8 μM) was preincubated for 15 min with eEF2·GTP (1 μM) and Phe-TC, Val-TC, and [$^3$H]-Lys-TC (all TCs present at 1.6 μM). The resulting complex was then reacted with puromycin.

### FVKM-tRNA$^{Met}$ bound in the P-site (Structure 13, POST-4)

80S·FVKM-IRES complex (0.8 μM) was preincubated for 15 min with eEF2·GTP (1 μM) and Phe-TC, Val-TC, Lys-TC and [$^{35}$S]-Met-TC (all TCs present at 1.6 μM). The resulting complex was then reacted with puromycin.

### FVK-tRNA$^{Lys}$ undergoing translocation (Structure 9 becoming Structure 10)

80S·FVKM-IRES complex (0.8 μM) was preincubated for 15 min with eEF2·GTP (1 μM) and Phe-TC and Val-TC, both present at 1.6 μM, yielding PheVal-tRNA$^{Val}$ bound in the P-site. This complex was then mixed for 1 min with ([$^3$H]-Lys-TC (1.6 μM) to form PheValLys-tRNA$^{Lys}$ bound in the A-site, which was then reacted with puromycin in the presence of additional added eEF2·GTP (final concentration 1 μM).

### FVKM-tRNA$^{Met}$ undergoing translocation (Structure 12 becoming Structure 13)

80S·FVKM-IRES complex (0.8 μM) was preincubated for 15 min with eEF2·GTP (1 μM) and Phe-TC, Val-TC, and Lys-TC (all TCs present at 1.6 μM), yielding PheValLys-tRNA$^{Lys}$ bound in the P-site. This complex was then mixed for 1 min with ([$^{35}$S]-Met-TC (1.6 μM) to form PheValLysMet-tRNA$^{Met}$ bound in the A-site, which was then reacted with puromycin in the presence of additional added eEF2·GTP (final concentration 1 μM).

## Acknowledgements

We thank Eric Jan for gifts of wt and variant CrPV Phe-IRES vectors and for helpful discussions.

## Additional information

### Funding

| Funder | Grant reference number | Author |
|---|---|---|
| National Institutes of Health | RO1GM 080376 | Haibo Zhang<br>Martin Y Ng<br>Yuanwei Chen<br>Barry S Cooperman |

The funder had no role in study design, data collection and interpretation, or the decision to submit the work for publication.

### Author contributions

HZ, Designed experiments and acquired, Analyzed and interpreted data; MYN, YC, Acquired, analyzed and interpreted data; BSC, Conceived and designed the experiments, Interpreted results, Wrote the manuscript

### Author ORCIDs

Barry S Cooperman, http://orcid.org/0000-0001-6989-9788

## Additional files

### Supplementary files

• Supplementary file 1. Initial coding sequences of variants used in this work.

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
