## [Decision Letter]

Thank you for submitting your work entitled "One, two, three, go – initiating polypeptide elongation in an IRES-dependent system" for consideration by *eLife*. Your article has been favorably evaluated by John Kuriyan (Senior editor) and three reviewers, one of whom, Alan Hinnebusch, is a member of our Board of Reviewing Editors.

The reviewers have discussed the reviews with one another and the Reviewing Editor has drafted this decision to help you prepare a revised submission.

Summary of work:

This paper conducts in vitro experiments using a purified system to study the rates of initial oligopeptide synthesis by an 80S·CrPV-IRES complex. The results suggest that the first two elongation cycles involved in producing a di- or tripeptide are slow and are limited by the translocation step, and that the rate of elongation accelerates following translocation of tripeptidyl-tRNA to the P-site (the second true translocation event). The IRES was suitably modified to replace the 1st Ala codon with a Phe codon, allowing fluorescence anisotropy measurements of the rate of proflavin-modified Phe-tRNA^Phe^ binding to either the A- or P-site, and to allow incorporation of labeled Met-tRNA^Met^ to monitor production of FM, FKM, FVKM, and FKVRQWLM synthesis. Sedimentation of complexes through sucrose gradients was used to assay stable binding of Met-tRNA or peptidyl-tRNAs containing labeled Met to the ribosome. The anisotropy results in Figure 3 suggest that EF2 shifts the equilibrium of the first pseudotranslocation step in which the IRES is moved in the decoding center to vacate the A-site, and thereby increases the rate of Phe-tRNA^Phe^ binding to the A site in a biphasic reaction. The sedimentation assay indicates that the second pseudotranslocation step, in which Phe-tRNA is translocated from the A to the P site is much slower than the combination of the first pseudotranslocation plus Phe-tRNA^Phe^ binding to the A site. They go on to use a rapid mixing and quench assay to measure rates of FM, FKM, and FKVM synthesis, with detection and quantification of product by thin layer electrophoresis (TLE), using the appropriate pre-formed substrates for addition of labeled Met to Phe-tRNA^Phe^ or the relevant peptidyl tRNA. Results in Figure 4 indicate that the rates of aa-tRNA binding to the A site and formation of a peptide bond are very similar for all 3 substrates that produce a di-, tri-, or tetrapeptide, and the calculated rates of translocation in these reactions indicate that the rates of translocation for the dipeptidyl- and tripeptidyl-tRNAs are slow and rate-limiting in forming the tri- and tetrapeptides, respectively. Using the sedimentation assay, they could also measure the very rapid rates of ternary complex binding to the A site in the 2nd and 3rd elongation cycles producing the tri- and tetrapeptides, but found that A-site bound TC or FM-tRNA was too unstable to capture for the intermediates 5-6 containing the IRES in the E site, which they speculate might involve an allosteric effect of the IRES in destabilizing A-site binding. By then measuring rates of puromycin reaction with either the tri- or tetrapeptidyl tRNAs either under conditions where prior translocation from the P-site to the A-site is required or where the peptidy-tRNAs are prebound to the A site, they deduce that the rate of the 3rd translocation of tetrapeptidyl-tRNA from the P to A site is very rapid compared to the 2nd or 3rd translocation events.

Essential revisions:

The following issues raised in the referees' comments shown below need to be addressed in the revised manuscript.

Ref. #1: All of the requests for clarification of reaction conditions or interpretations of results should be addressed with appropriate revisions of text.

Ref.#2: The request for complete details about fitting the kinetic data in comment #1 should be honored.

As described in comment #2, the need for modifying the IRES sequence should be more explicitly stated in Materials and methods, as it would have been technically feasible to carry out all experiments with the WT sequence. In addition, it is important to compare the rates of IRES-mediated translation given by the native sequence versus your modified IRES sequence either in a cell-free extract or in your reconstituted system.

To respond to points 3. & 4., it would be sufficient to revise the text to remove statements about allostery; for point 5, to explain why you monitored the puromycin reaction as you did in Figure 5. For points 6-10, clarify the relevant issues with the appropriate revisions of text, and provide the requested control to rule out Met-tRNA binding to 30S subunits.

Ref. #3: Points 1.-3 concern your conclusion that eEF2 does not stimulate the rate of TC binding in step 2 and rather only shifts the equilibrium in step 1 towards structure 2, which is at odds with other published work. As this is a hypothetical interpretation of the biphasic kinetics in Figure 3, it is important that you attempt to provide evidence for this deduced effect of eEF2 on the equilibrium of step 1 using toe-printing analysis of the complexes. It is also necessary to justify the statement that k-1 is much smaller than k2, which is not self-evident, and explain how it could be that eEF2 would shift the equilibrium to structure 2 without altering the t_1/2_ of step 1. You must also discuss the discrepancy with the previous results, particularly those of Ruehle et al.

You should clarify the issues raised in points 4-6 with the appropriate revisions of text.

Original reviews:

Reviewer #1:

The results are significant in providing strong evidence that the two pseudotranslocation steps (steps 1 and 3 in Figure 2) as well as the two conventional translocation steps (steps 6 and 9) involved in producing a tripeptide and translocating it to the P site occur slowly and are rate-limiting for the first two elongation cycles of CrPV IRES-directed translation, and that the next translocation step occurs much more rapidly and does not appear to be rate-limiting for production of longer polypeptides. The slow rate of the 2nd translocation step is surprising because the IRES should no longer occupy the decoding sites, so they may be detecting a novel intermediate with the IRES still bound to the ribosome outside of the decoding center and affecting a ribosome conformational change involved in translocation. The results are also significant in providing evidence against the previous conclusion by others that EF2 is required for the first pseudotranslocation that moves the IRES out of the A site (step 1) to permit recruiting the first TC to the A-site, as here they found no effect of EF2 on the rate of this reaction when it is monitored directly by fluorescence anisotropy measurements as opposed to stable TC binding to the P site assayed by sedimentation, which occurs only after the second pseudotranslocation reaction (step 3). Finally, the results using the CrPV IRES are valuable in laying the groundwork for studying eukaryotic elongation in a simplified purified system without the need for initiation factors to form the first peptide bound. There are a number of instances where the reaction conditions or interpretations of results need to be more carefully described, as follows:

1) In the second paragraph of the subsection “Rates of oligopeptide formation and Met-tRNA^Met^ cosedimentation”: This section is very confusing as the composition of the aa-tRNA or peptidyl-tRNA species in question, and whether it is bound to the A or P sites, is generally unclear. Particularly, in the clause "…These rate differentials provide a clear indication that peptidyl- Met-tRNA^Met^ and Met-tRNA^Met^ bound to the A-site (Structures 8, 9 and 11, 12), as well as peptidyl-Met-tRNA^Met^ binding to the P-site (Structures 10 and 13), efficiently…", the clarity could be improved by mentioning the specific peptidyl-tRNAs by their amino acid compositions for each of the structures, as "peptidyl-Met-tRNA^Met^" is imprecise nomenclature.

2) In the subsection “Rates of oligopeptide formation and Met-tRNA^Met^ cosedimentation”: Also, it's not perfectly clear whether the instability of Met-tRNA binding to the A-site observed for structure 3 and 5, and FM-tRNA to structure 6 using the co-sedimentation assay is unusual; whereas stable binding of both Met-tRNA and peptidyl-tRNA to the A-site in structures 8, 9, 11-12 is typical of A-site-bound ligands in conventional (non-IRES) elongation complexes; and this distinction needs to be carefully spelled out.

3) In the second paragraph of the subsection “Translocation of tetrapeptidyl-tRNA (Step 12) is much more rapid than of tripeptidyl-tRNA (Step 9)”: requires citation of a figure for puromycin synthesis rates.

4) In the second paragraph of the subsection “Translocation of tetrapeptidyl-tRNA (Step 12) is much more rapid than of tripeptidyl-tRNA (Step 9)” and Figure 5: more detail is required about how the experiment in Figure 5 was done in terms of starting reactants.

5) In the last paragraph of Results: more detail is required to understand how binding of peptidyl tRNA harboring labeled Met is distinguished from Met-tRNA binding to the A-site in these assays.

6) Error estimates have not been defined in any of the figures or tables and this needs to be rectified.

Reviewer #2:

The paper by Cooperman et al. provides a first kinetic description of the initial steps of translation by eukaryotic ribosomes after the initiation at an IRES. There are very few papers on mechanistic aspects of eukaryotic translation in general and the present work provides very interesting, significant insights. It is a real challenge to obtain clean kinetic data for the eukaryotic system and this paper provides valuable information on the kinetic properties of eukaryotic ribosomes. The technical quality of the data is high (I particularly like the quality of the stopped-flow traces, they are really excellent) and the conclusions are warranted. My only reservation is that many technical aspects are not clearly described. Given that this paper delineates new kinetic approaches to study eukaryotic translation, it is particularly important to provide a very detailed description of methods. This really has to be carefully revised. I personally do not like the title, because it is not informative at all; a more rigorous title would be better.

Detailed comments:

1) Fitting of the kinetic data in this paper is mostly non-exponential and would require fitting to a model by numerical integration. This part is completely missing in Materials and methods. This must be described in all details required to understand and reproduce the calculations.

2) Among all potential criticism of the experimental system (Discussion, third paragraph), I am particularly worried about the altered coding sequence. It is not clear why the native Ala codon was replaced by Phe, as unlabeled amino acids are used anyway and the tRNAs are prepared by a method which should readily yield tRNA(Ala), i.e. one could label tRNA(Ala) with Prf (Kothe and Rodnina, 2007). This means that in principle one could use the native sequence. Furthermore, the measurements and rate determinations would be even more straightforward if [14C] and [3H]-labeled amino acids were used instead of [^35^S]Met; in this case one could use the native sequence. The need for modifying the sequence should be more explicitly stated in Materials and methods. Clearly, the whole set of experiments cannot be repeated with Ala as a 1st amino acid; however, it would be highly desirable to compare IRES-mediated translation with a native sequence and a modified sequence in the cell-free translation system (commercial or reconstituted from components).

3) The "instability" of aa-tRNA binding in A site is not likely to be caused by the dissociation of aa-tRNA from the A site prior to peptide bond formation. As authors show, peptide bond formation is rapid (Table 1); so as soon as aa-tRNA binds, it will be rapidly incorporated into peptide. Instead, peptidyl-tRNA tends to drop-off easily (see Semenkov 2000 and Konevega 2004). Although this dissociation is relatively slow, it normally explains the drop-off during centrifugation (which takes minutes to hours). In any case, there is no evidence for the allosteric interactions between the A and E sites. The statements on the allosteric interplay should be removed, as they only weaken the paper (subsection “Rates of oligopeptide formation and Met-tRNA^Met^ cosedimentation”, third paragraph) (By the way, the so-called "allosteric effect" is simply caused by the presence of deacylated tRNA in aa-tRNA preps, which chases the labeled tRNA from the E site. There are several other groups, in addition to Rachel Green, who provided strong evidence against the model).

4) The instability issues addressed in point 3 could be overcome by using nitrocellulose filtration assays, which are really very commonly used to study ribosome studies. Are there any technical issues which preclude the use of nitrocellulose filter in this case? If yes, this should be described in Materials and methods.

5) The authors use puromycin reaction to identify the P-site position of the peptidyl-tRNA, which is a very reliable and well-established method. It is therefore surprising that the authors did not use the time-resolved puromycin assays (which they have established for the prokaryotic ribosomes) to determine the rates of translocation in a more direct fashion than described in Table 1? If there is some technical issue specific for eukaryotic ribosomes, it would be important to indicate this in Materials and methods.

6) Discussion: "Further work will be required to determine how many cycles of elongation are required before any retarding influence of bound IRES is completely eliminated": I am confused here. Table 1 shows that k12 is very fast, i.e. the 5th round is already rapid. This should be clarified.

7) In the subsection “Complex preparations. TCs and various 80S·IRES complexes”. 30S subunits were added to the centrifugation assays as a carrier. However, 30S subunits can bind [^35^S]Met (at least to some extent) even in the absence of the mRNA. Are there controls for that?

8) Figure 1 legend. "in the both cases" – it is not clear which two cases are meant. It is also not clear why the incubation with EF2 is so long (1-2 hours) – is it really necessary? I guess the authors just wanted to be on the safe side, but this has to be clearly stated in Materials and methods.

9) Figure 4 legend; "or just [^35^S]Met-TC”. My expectation is that in the absence of the preceding Lys-TC Met should not be incorporated at all. This has to be clarified.

10) In the subsection “Kinetic measurements”, "averages of 2-4 independent determinations" – do the authors mean technical replicates or independent experiments (i.e. biological replicated)?

Reviewer #2 (Additional data files and statistical comments):

The statistical information is appropriate except for the clarification needed in the subsection “Kinetic measurements” (point 10 in the review).

Reviewer #3:

Zhang and colleagues have investigated the effect of the CrPV IRES RNA on the first rounds of translation elongation including the initial pseudo-translocation steps. They have performed for the first time experiments to determine the rates of the various main steps of the elongation cycles. They show that the IRES initial significantly slows down the translocation steps are down until the tetrapeptide stage is reached. The CrPV is an important minimal model system for translation initiation in eukaryotes The results are novel and interesting. However, the taking into account the following points is recommended before the paper can be published.

1) According to Figure 3, blue curve, the binding of tRNA to the A-site (structure 2 in the nomenclature of the authors) reaches nearly the same level with or without eEF2. However, in previous experiments binding of tRNA was dependent on eEF2 or at least significantly increased (Yamamoto et al., 2007; Fernandez et al., 2014; Ruehle et al., 2015). The experiments by Ruehle et al. have been done by colocalization using fluorescence. The discrepancy should be discussed. Furthermore, the authors should perform experiments to establish the nature of the complex in a more direct manner, e.g. by toeprinting of the complex.

2) The authors state that k-1 is much smaller than k2 and was assumed to be negligible. How has this been determined? In fact, a toeprint signal indicative of translocation cannot be obtained by incubation of binary 80S-IRES complexes with eEF2 alone, but requires an A-site ligand, too (Jan et al., 2003; Muhs et al., 2015). This suggest that the translocated complex (structure 2) is kinetically labile and contradicts the present statement that the back-translocation is negligible. Again, the authors should do experiments, e.g. toeprinting, to provide additional evidence that pre-incubation with eEF2-GTP shifts the equilibrium between structures 1 and 2 from 95:5 to 50:50 as stated.

3) The explanation about the effect of eEF2 on the equilibrium between structure 1 and 2 (Discussion) is curious. They propose that eEF2 shifts the equilibrium towards structure 2 at the same time do not measure a significant effect on the t_1/2_ of step 1. How this is possible? Do the authors propose that k-1 is becoming smaller, i.e. that eEF2 slows down back-translocation without accelerating forward translocation?

4) Can the authors rule out that back-translocation or peptidyl-tRNA drop off influences the following steps towards tetrapeptide synthesis?

Additional points:

5) The first sentence of the paper "Initiation of protein synthesis in eukaryotic cells proceeds via two well-established pathways" is to some extent misleading. As the authors acknowledge later factor requirement of different IRES RNAs is diverse and the structure and mechanism of different IRES RNAs is also different. So there are several pathways leading to internal initiation.

6) The peptide coding sequence is not really part of the IRES. Therefore, it is misleading to talk about mutant IRES, when the changes are in the ORF.

7) At the beginning of the Results part the authors should specify in which system they are working.

8) The green line is not mentioned in the legend to Figure 3.

9) A-site/E-site allostery has been recently reported by Ferguson et al., 2015, Mol Cell for human 80S ribosomes. This is relevant for the respective discussion in the third paragraph of the subsection “Rates of oligopeptide formation and Met-tRNA^Met^ cosedimentation”.

[Editors' note: further revisions were requested prior to acceptance, as described below.]

Thank you for resubmitting your work entitled "Kinetics of initiating polypeptide elongation in an IRES-dependent system" for further consideration at *eLife*. Your revised article has been favorably evaluated by John Kuriyan (Senior editor), a Reviewing editor, and two reviewers.

The manuscript has been improved but there are some remaining issues raised by Reviewer #3 that need to be addressed before acceptance. In considering these new comments, Reviewer #2 also asks that you carefully review the Materials and methods to insure that they contain all of the critical information that would be required to duplicate the work.

Reviewer #2:

The authors have significantly improved the manuscript by satisfactorily addressing the comments and concerns of the reviewers. The story they prevent extends our knowledge on the function IRES-dependent systems and can be published in *eLife*.

Reviewer #3:

In the revised version Zhang and colleagues have addressed many points raised by the reviewers and have improved their paper. However, there are still some points where discussion should be extended.

1) The authors now confirm that the effect of eEF2 is to inhibit k-1 and write that this is "consistent with its role as a translocase". However, the accepted role of EF-G/eEF2 is to accelerate translocation, not to inhibit back-translocation. This warrants additional discussion.

2) The authors have pre-incubated 80S-FVKM-IRES complexes with eEF2-GTP for 1 – 2 hr. How they can rule out that GTP consumption has an impact on the experiment? Is it possible that accumulation of eEF2-GDP during pre-incubation results in a significant fraction of 80S FVKM-IRES-eEF2 complexes (between Structures 1 and 2) to slow down k2.

3) The authors write that the anisotropy assay in Figure 3 reports on formation of Structure 3 from 1. However, as the co-sedimentation assay reports, during the measurement time there should be also formation of Structure 4 at later time.

4) The authors state that A-site-bound Phe-tRNA^Phe^ is labile. Can they estimate the off-rate k-2. Is it valid to neglect k-2 for numerical integration?

---

## [Author Response]

*Essential revisions:*

*The following issues raised in the referees' comments shown below need to be addressed in the revised manuscript.*

Ref. #1: All of the requests for clarification of reaction conditions or interpretations of results should be addressed with appropriate revisions of text.

This has been addressed.

Ref.#2: The request for complete details about fitting the kinetic data in comment #1 should be honored.

This has been addressed.

*As described in comment #2, the need for modifying the IRES sequence should be more explicitly stated in Materials and methods, as it would have been technically feasible to carry out all experiments with the WT sequence.*

This was simply a matter of convenience, as now stated explicitly in the first paragraph of the Materials and methods. When we started this work tRNA^Phe^ was available to us and the tRNA^Ala^ acceptor was not, and Eric Jan told us that the Phe codon UUC could be substituted for the Ala GCU codon with little or no functional consequence.

In addition, it is important to compare the rates of IRES-mediated translation given by the native sequence versus your modified IRES sequence either in a cell-free extract or in your reconstituted system.

This has been addressed by addition of Figure 1—figure supplement 1.

To respond to points 3. & 4., it would be sufficient to revise the text to remove statements about allostery; for point 5, to explain why you monitored the puromycin reaction as you did in Figure 5. For points 6-10, clarify the relevant issues with the appropriate revisions of text, and provide the requested control to rule out Met-tRNA binding to 30S subunits.

These points have been addressed, but, as explained in response to Reviewer #2, point #3, we would prefer to leave the allosteric discussion in the text.

Ref. #3: Points 1.-3 concern your conclusion that eEF2 does not stimulate the rate of TC binding in step 2 and rather only shifts the equilibrium in step 1 towards structure 2, which is at odds with other published work. As this is a hypothetical interpretation of the biphasic kinetics in Figure 3, it is important that you attempt to provide evidence for this deduced effect of eEF2 on the equilibrium of step 1 using toe-printing analysis of the complexes. It is also necessary to justify the statement that k-1 is much smaller than k2, which is not self-evident, and explain how it could be that eEF2 would shift the equilibrium to structure 2 without altering the t_1/2_ of step 1. You must also discuss the discrepancy with the previous results, particularly those of Ruehle et al.

We disagree with the request for a toeprinting experiment, for reasons elaborated in response to Reviewer 3’s comments #s 1 – 3. However, we have carried out a more complete analysis of our kinetic data, resulting in the addition of a new table (revised Table 1) and a more cogent presentation of t_1/2_ values for Steps 1 and 2 in revised Table 2 (formerly Table 1). We are indebted to Reviewer 3 for raising this point. We also rebut the statement that there is a discrepancy between our present results and those of Ruehle et al.

You should clarify the issues raised in points 4-6 with the appropriate revisions of text.

We have addressed points 4 and 6 in our response. We disagree with point 5, as stated in our detailed response.

*Original reviews:*

*Reviewer #1:*

*The results are significant in providing strong evidence that the two pseudotranslocation steps (steps 1 and 3 in Figure 2) as well as the two conventional translocation steps (steps 6 and 9) involved in producing a tripeptide and translocating it to the P site occur slowly and are rate-limiting for the first two elongation cycles of CrPV IRES-directed translation, and that the next translocation step occurs much more rapidly and does not appear to be rate-limiting for production of longer polypeptides. The slow rate of the 2nd translocation step is surprising because the IRES should no longer occupy the decoding sites, so they may be detecting a novel intermediate with the IRES still bound to the ribosome outside of the decoding center and affecting a ribosome conformational change involved in translocation. The results are also significant in providing evidence against the previous conclusion by others that EF2 is required for the first pseudotranslocation that moves the IRES out of the A site (step 1) to permit recruiting the first TC to the A-site, as here they found no effect of EF2 on the rate of this reaction when it is monitored directly by fluorescence anisotropy measurements as opposed to stable TC binding to the P site assayed by sedimentation, which occurs only after the second pseudotranslocation reaction (step 3). Finally, the results using the CrPV IRES are valuable in laying the groundwork for studying eukaryotic elongation in a simplified purified system without the need for initiation factors to form the first peptide bound. There are a number of instances where the reaction conditions or interpretations of results need to be more carefully described, as follows:*

*1) In the second paragraph of the subsection “Rates of oligopeptide formation and Met-tRNA^Met^ cosedimentation”: This section is very confusing as the composition of the aa-tRNA or peptidyl-tRNA species in question, and whether it is bound to the A or P sites, is generally unclear. Particularly, in the clause "…These rate differentials provide a clear indication that peptidyl- Met-tRNA^Met^ and Met-tRNA^Met^ bound to the A-site (Structures 8, 9 and 11, 12), as well as peptidyl-Met-tRNA^Me^t binding to the P-site (Structures 10 and 13), efficiently…", the clarity could be improved by mentioning the specific peptidyl-tRNAs by their amino acid compositions for each of the structures, as "peptidyl-Met-tRNA^Met^" is imprecise nomenclature.*

The requested clarifications have been made, both in the Results and Experimental Sections.

*2) In the subsection “Rates of oligopeptide formation and Met-tRNA^Met^ cosedimentation”: Also, it's not perfectly clear whether the instability of Met-tRNA binding to the A-site observed for structure 3 and 5, and FM-tRNA to structure 6 using the co-sedimentation assay is unusual; whereas stable binding of both Met-tRNA and peptidyl-tRNA to the A-site in structures 8, 9, 11-12 is typical of A-site-bound ligands in conventional (non-IRES) elongation complexes; and this distinction needs to be carefully spelled out.*

The requested distinction has been made. See subsection “Rates of oligopeptide formation and Met-tRNA^Met^ cosedimentation”, end of second paragraph and start of the third paragraph.

*3) In the second paragraph of the subsection “Translocation of tetrapeptidyl-tRNA (Step 12) is much more rapid than of tripeptidyl-tRNA (Step 9)”: requires citation of a figure for puromycin synthesis rates.*

The references for the reactivity of P-site bound aminoacyl-tRNA with puromycin in eukaryotic ribosomes were originally provided (Lorsch and Herschlag, 1999 and Ioannu et al., 1997). In writing the revised version we became aware of a paper published in late 2015 by Ferguson et al., which also provided a pertinent result, and this reference was also added. In addition, in considering this comment we realized that there were no reliable data extant for the reactivity of A-site bound aminoacyl-tRNA with puromycin in eukaryotic ribosomes. Accordingly, we performed the relevant experiment for A-site bound PheValLys-tRNA^Lys^, demonstrating that A-site reactivity is approximately 20-fold less than P-site reactivity. This result, which has been added to Figure 5 with a description in the third paragraph of the subsection “Translocation of tetrapeptidyl-tRNA (Step 12) is much more rapid than of 182 tripeptidyl-tRNA (Step 9)”, fully supports our original conclusion, based on results presented in our initial submission, that translocation of PheValLys-tRNA^Lys^ is much slower than translocation of PheValLysMet-tRNA^Met^. We also added two sentences (in the aforementioned paragraph) explicitly comparing A-site vs. P-site reactivity with puromycin in prokaryotic and eukaryotic ribosomes.

*4) In the second paragraph of the subsection “Translocation of tetrapeptidyl-tRNA (Step 12) is much more rapid than of tripeptidyl-tRNA (Step 9)” and Figure 5: more detail is required about how the experiment in Figure 5 was done in terms of starting reactants.*

More detail was added to the description of the puromycin reaction in the subsection entitled “Rates of puromycin adduct formation” as requested. In addition, the reference to Figure 4 in the original version of this section was corrected to Figure 5.

*5) In the last paragraph of Results: more detail is required to understand how binding of peptidyl tRNA harboring labeled Met is distinguished from Met-tRNA binding to the A-site in these assays.*

A sentence justifying the use of the cosedimentation assay to measure octapeptide formation involving peptide transfer to ribosome-bound Met-tRNA^Met^ has been added to the subsection entitled “Octapeptide” in the Experimental section. We also added Figure 5—figure supplement 1 that confirms octapeptide synthesis.

6) Error estimates have not been defined in any of the figures or tables and this needs to be rectified.

In the subsection labeled “Kinetic measurements” the last sentence has been amended to read “Error bars in figures are shown as average deviations.” In addition, several additions were made to revised Table 2 (formerly Table 1), including a full description of the error ranges, added as footnotes a. and d., as well as the measured t_1/2_ values for the combined steps 4-8 and 7-11.

*Reviewer #2:*

The paper by Cooperman et al. provides a first kinetic description of the initial steps of translation by eukaryotic ribosomes after the initiation at an IRES. There are very few papers on mechanistic aspects of eukaryotic translation in general and the present work provides very interesting, significant insights. It is a real challenge to obtain clean kinetic data for the eukaryotic system and this paper provides valuable information on the kinetic properties of eukaryotic ribosomes. The technical quality of the data is high (I particularly like the quality of the stopped-flow traces, they are really excellent) and the conclusions are warranted. My only reservation is that many technical aspects are not clearly described. Given that this paper delineates new kinetic approaches to study eukaryotic translation, it is particularly important to provide a very detailed description of methods. This really has to be carefully revised. I personally do not like the title, because it is not informative at all; a more rigorous title would be better.

The title has been changed as requested.

*Detailed comments:*

*1) Fitting of the kinetic data in this paper is mostly non-exponential and would require fitting to a model by numerical integration. This part is completely missing in Materials and methods. This must be described in all details required to understand and reproduce the calculations.*

Only the results presented in Figure 3 and Figure 3—figure supplement 2 were fit using the numerical integration program Scientist. Although Scientist was mentioned in the Figure 3 legend of the original version, this point is now emphasized in the revised version by adding the phrase “the numerical integration program” to the last line of the figure legend.

*2) Among all potential criticism of the experimental system (Discussion, third paragraph), I am particularly worried about the altered coding sequence. It is not clear why the native Ala codon was replaced by Phe, as unlabeled amino acids are used anyway and the tRNAs are prepared by a method which should readily yield tRNA(Ala), i.e. one could label tRNA(Ala) with Prf (Kothe and Rodnina, 2007). This means that in principle one could use the native sequence. Furthermore, the measurements and rate determinations would be even more straightforward if [14C] and [3H]-labeled amino acids were used instead of [^35^S]Met; in this case one could use the native sequence. The need for modifying the sequence should be more explicitly stated in Materials and methods. Clearly, the whole set of experiments cannot be repeated with Ala as a 1st amino acid; however, it would be highly desirable to compare IRES-mediated translation with a native sequence and a modified sequence in the cell-free translation system (commercial or reconstituted from components).*

An experiment demonstrating that replacement of the initial Ala codon GCU by UUC encoding Phe has little effect on active luciferase expression has been added to the text as Figure 1—figure supplement 1 and referenced in the text (1^st^ para in Results), with experimental details provided (subsections “In Vitro Transcription” and “Luciferase assay”).

*3) The "instability" of aa-tRNA binding in A site is not likely to be caused by the dissociation of aa-tRNA from the A site prior to peptide bond formation. As authors show, peptide bond formation is rapid (Table 1); so as soon as aa-tRNA binds, it will be rapidly incorporated into peptide. Instead, peptidyl-tRNA tends to drop-off easily (see Semenkov 2000 and Konevega 2004). Although this dissociation is relatively slow, it normally explains the drop-off during centrifugation (which takes minutes to hours). In any case, there is no evidence for the allosteric interactions between the A and E sites. The statements on the allosteric interplay should be removed, as they only weaken the paper (subsection “Rates of oligopeptide formation and Met-tRNA^Met^ cosedimentation”, third paragraph) (By the way, the so-called "allosteric effect" is simply caused by the presence of deacylated tRNA in aa-tRNA preps, which chases the labeled tRNA from the E site. There are several other groups, in addition to Rachel Green, who provided strong evidence against the model).*

There are two points raised here. We and the reviewer are in agreement that PheMet-tRNA^Met^ is not bound stably to the A-site. However, we also think it possible that Met-tRNA^Met^ bound to the A-site may also be bound unstably. This is based on our observation of a large difference in the relative stoichiometries measured by the co-sedimentation and peptide synthesis assays at 3 s and 10 s after Met-TC addition (Figure 4). Based on our estimated t_1/2_ values for the peptide formation steps 8 and 11 (4 s and 7s, respectively), we would expect both PheMet-tRNA^Met^ and Met-tRNA^Met^ to be bound to the ribosome within 10 s after Met-TC addition, and so cannot rule out that Met-tRNA^Met^ is bound unstably to the A-site following step 4. To reflect this uncertainty, we have changed the wording of the relevant sentence in the text to read:

“This indicates that PheMet-tRNA^Met^, and possibly Met-tRNA^Met^ as well, are not bound stably to the ribosome in Structures 5 and6”.

As to the second point, there are indeed other groups than Green et al. who have argued against an allosteric A-site/E-site effect, in particular Wintermeyer, Rodnina and their co-workers, and an appropriate reference (Semenkov et al., 1996) has been added to the text. However, there remains a need to explain the very evident differences we observe between the relative stoichiometries measured by the co-sedimentation and peptide synthesis assays in the first (Figure 4), second (Figure 4), and third (Figure 4) peptide elongation steps. As stated in the revised text we think that:

“It is possible that the lability of the A-site tRNAs in structures 5and6is due to IRES binding to the E-site, which is absent in structures 8, 9and 11, 12**”**.

So then the question becomes why would a putative IRES binding to the E-site cause this difference in behavior. Above we have mentioned studies arguing against A-site:E-site allosteric interaction, but there are also studies that support the notion, most notably by Nierhaus (Nierhaus 1990), but also by ourselves (Chen et al., 2011), and most recently, by Ferguson et al. (2015), the latter notably based on work with eukaryotic ribosomes. Given this background, we think it is permissible for us to speculate that the lability of A-site tRNAs in structures 5and6:

“may reflect an allosteric A-site: E-site interaction”.

(see also Reviewer #3, point #9). This said, we are willing to withdraw the latter phrase and the references to the A-site:E-site controversy if the editors object to their inclusion.

*4) The instability issues addressed in point 3 could be overcome by using nitrocellulose filtration assays, which are really very commonly used to study ribosome studies. Are there any technical issues which preclude the use of nitrocellulose filter in this case? If yes, this should be described in Materials and methods.*

We did not use nitrocellulose filter assays, and do not think that such assays are required for this paper.

*5) The authors use puromycin reaction to identify the P-site position of the peptidyl-tRNA, which is a very reliable and well-established method. It is therefore surprising that the authors did not use the time-resolved puromycin assays (which they have established for the prokaryotic ribosomes) to determine the rates of translocation in a more direct fashion than described in Table 1? If there is some technical issue specific for eukaryotic ribosomes, it would be important to indicate this in Materials and methods.*

There are technical issues. The rate of puromycin reaction with peptidyl-tRNA bound in the P-site of the eukaryotic ribosome is much slower (several hundred-fold) than the corresponding rate in the prokaryotic ribosome, so that only if the translocation rate is even slower (as in the case of the tripeptidyl-tRNA) is it possible to use the time-resolved puromycin assay to get information about the translocation rate. Second, we were not confident of the stability of A-site bound peptidyl-tRNA toward co-sedimentation and so decided to generate A-site bound peptidyl-tRNA in situ prior to puromycin addition. More detail about the puromycin reaction has been added to both the Results (subsection “Translocation of tetrapeptidyl-tRNA (Step 12) is much more rapid than of tripeptidyl-tRNA (Step 9)”) and Experimental (subsection “Kinetic measurements”) sections of the revised manuscript (see also response to Reviewer 1’s comments #s 3 and 4).

*6) Discussion: "Further work will be required to determine how many cycles of elongation are required before any retarding influence of bound IRES is completely eliminated": I am confused here. Table 1 shows that k12 is very fast, i.e. the 5th round is already rapid. This should be clarified.*

A phrase has been added to the relevant sentence to make this point clear.

*7) In the subsection “Complex preparations. TCs and various 80S·IRES complexes”. 30S subunits were added to the centrifugation assays as a carrier. However, 30S subunits can bind [^35^S]Met (at least to some extent) even in the absence of the mRNA. Are there controls for that?*

Yes. Co-sedimentation due to binding to 30S subunits is negligible. This point has been addressed by adding both an appropriate sentence to the subsection in Materials and methods entitled “Complex Preparations”and Figure 4—figure supplement 1 which provides illustrative data.

*8) Figure 1 legend. "in the both cases" – it is not clear which two cases are meant. It is also not clear why the incubation with EF2 is so long (1-2 hours) – is it really necessary? I guess the authors just wanted to be on the safe side, but this has to be clearly stated in Materials and methods.*

The reviewer is referring to the Figure 3 legend and is correct about the reasoning. A sentence has been added to the legend make this point explicit.

*9) Figure 4 legend; "or just [^35^S]Met-TC”. My expectation is that in the absence of the preceding Lys-TC Met should not be incorporated at all. This has to be clarified.*

Clarifying text has been added. The basic point is that, in 4C, the kinetics of tripeptide synthesis was followed for either 5 steps (Structure 4 converted to Structure 9) or two steps (Structure 7 converted to Structure 9). Similarly, in 4D the kinetics of tetrapeptide synthesis was followed for either 5 steps (Structure 7 converted to Structure 12) or two steps (Structure 10 converted to Structure 12).

10) In the subsection “Kinetic measurements”, "averages of 2-4 independent determinations" – do the authors mean technical replicates or independent experiments (i.e. biological replicated)?

Clarifying language on this point has been added to the subsection “Kinetic measurements”.

*Reviewer #3:*

*Zhang and colleagues have investigated the effect of the CrPV IRES RNA on the first rounds of translation elongation including the initial pseudo-translocation steps. They have performed for the first time experiments to determine the rates of the various main steps of the elongation cycles. They show that the IRES initial significantly slows down the translocation steps are down until the tetrapeptide stage is reached. The CrPV is an important minimal model system for translation initiation in eukaryotes The results are novel and interesting. However, the taking into account the following points is recommended before the paper can be published.*

*1) According to Figure 3, blue curve, the binding of tRNA to the A-site (structure 2 in the nomenclature of the authors) reaches nearly the same level with or without eEF2. However, in previous experiments binding of tRNA was dependent on eEF2 or at least significantly increased (Yamamoto et al., 2007; Fernandez et al., 2014; Ruehle et al., 2015). The experiments by Ruehle et al. have been done by colocalization using fluorescence. The discrepancy should be discussed. Furthermore, the authors should perform experiments to establish the nature of the complex in a more direct manner, e.g. by toeprinting of the complex.*

The reviewer makes two suggestions for revising the manuscript in this point, and we disagree with both.

There is no conflict with the Ruehle et al. paper, on which we are co-authors. The colocalization result is reported in Figure 6 of Ruehle et al. and utilizes single molecule observation. The relevant experimental section describing this experiment in the Ruehle et al. paper reads as follows:

“The 80S-IRES ribosome complex was incubated in the flowcell for 5 min and components that remained untethered to the surface of the microfluidic flowcell at the conclusion of the 5 min were washed out of the flowcell using an imaging buffer…”.

This washing procedure has the effect of removing the labile A-site bound Phe-tRNA^Phe^, accounting for the much lower stoichiometry observed in the absence of added eEF2. This point is addressed directly in Ruehle et al. in the section entitled “Loop 3 facilitates eEF2’s ability to translocate ac-tRNA on IGR IRES-80S ribosome complexes”:

“To examine eEF1A-dependent ac-tRNA delivery, we assembled TC with Phe353

tRNA^Phe^(Cy5)+eEF1A+GTP and delivered this to the immobilized IRES-80S complexes without eEF2. Compared to the reactions lacking eEF1A, both IRESs show increased and similar ac-tRNA occupancies (WT: 17.9 ± 4.8%, Δ3: 20.8 ± 5.4%). These data initially seem at odds with the anisotropy data in which eEF2-independent ac-tRNA association with 80S-WT IRES ribosome complexes is much greater than complexes with Δ3. This apparent discrepancy is likely due to the fact that anisotropy data are obtained under equilibrium conditions where transient interactions are observed, whereas the single-molecule fluorescence data are collected after the flowcell is flushed and thus only show stable long-lived association.”

The second suggestion requests that we clarify “the nature of the complex” through use of a toeprinting experiment. We are not completely clear to which complex the Reviewer is referring: Structure 2 with the A-site empty in either the absence or presence of eEF2; or Structure 3, with aa-tRNA bound in a labile manner in the A-site. Of these three possibilities, we consider two to be noncontroversial. In the absence of eEF2 and Phe-TC, our results suggest that the equilibrium between Structures 1 and 2 strongly favors 1, in agreementwith the results of others. With respect to Structure 3, the results presented in the Ruehle et al. paper show the tRNA binding to be enhanced for cognate vs. non-cognate tRNA, supporting the structure shown.

Where our results may appear to be at odds with published results, likely prompting the Reviewer’s suggestion, are for Structure 2 in the presence of added eEF2·GTP. We address this point directly in the revised text (Discussion, third paragraph). The basic conclusion is that the reaction conditions employed in the toeprinting assay make it an unreliable method for measuring eEF2·GTP effects on the 1 to 2equilibrium position, in agreement with an earlier suggestion of Muhs et al. (2015) (subsection “Di-, Tri- and Tetrapeptide.”), which has recently been supported by results of Petrov et al. (2016). Because of this, we do not agree with the reviewer that a toeprinting experiment is likely “to establish the nature of the complex in a more direct manner”.

*2) The authors state that k-1 is much smaller than k2 and was assumed to be negligible. How has this been determined? In fact, a toeprint signal indicative of translocation cannot be obtained by incubation of binary 80S-IRES complexes with eEF2 alone, but requires an A-site ligand, too (Jan et al., 2003; Muhs et al., 2015). This suggest that the translocated complex (structure 2) is kinetically labile and contradicts the present statement that the back-translocation is negligible. Again, the authors should do experiments, e.g. toeprinting, to provide additional evidence that pre-incubation with eEF2-GTP shifts the equilibrium between structures 1 and 2 from 95:5 to 50:50 as stated.*

*3) The explanation about the effect of eEF2 on the equilibrium between structure 1 and 2 (Discussion) is curious. They propose that eEF2 shifts the equilibrium towards structure 2 at the same time do not measure a significant effect on the t_1/2_ of step 1. How this is possible? Do the authors propose that k-1 is becoming smaller, i.e. that eEF2 slows down back-translocation without accelerating forward translocation?*

These two valid points are linked, and we thank the reviewer for raising them. To address them we fit the anisotropy results presented in Figure 3 to the Scheme presented in Figure 2 for Steps 1 and 2, yielding the apparent kinetic constants collected in new Table 1. The text has been modified (subsection “Rates of Phe-TC binding to the 80S·IRES complex: Steps 1-3, structures 1 – 4”, second paragraph) to include these results. The values in Table 1 show that, as expected, in the absence of eEF2·GTP, *k*_-1_ >> *k*_1_, whereas in the presence of eEF2·GTP, *k*_-1_ ~ *k*_1_. This change results from the effects of eEF2·GTP in strongly decreasing *k*_-1_ (~ 50-fold) while only modestly decreasing *k*_1_ (~ 2-fold). The rate constants in Table 1 are then used to calculate t_1/2_ values for Steps 1 and 2 in Table 2, as described in footnotes b and c.

4) Can the authors rule out that back-translocation or peptidyl-tRNA drop off influences the following steps towards tetrapeptide synthesis?

We do not think that peptidyl-tRNA drop off is a major issue, because the yield of peptide per ribosome is constant for di-, tri- and tetrapeptide. However, we acknowledge that many of the details of the process, including possible back translocation during oligopeptide synthesis, remain to be elucidated.

*Additional points:*

*5) The first sentence of the paper "Initiation of protein synthesis in eukaryotic cells proceeds via two well-established pathways" is to some extent misleading. As the authors acknowledge later factor requirement of different IRES RNAs is diverse and the structure and mechanism of different IRES RNAs is also different. So there are several pathways leading to internal initiation.*

We disagree with the reviewer. Our division of initiation of protein synthesis via two pathways is a common way of introducing the subject and follows the presentation in a review article we cite (Jackson et al., 2010). The qualification that IRES-initiated synthesis can take place in diverse ways comes three lines later in the opening paragraph, so we are hardly neglecting this point.

*6) The peptide coding sequence is not really part of the IRES. Therefore, it is misleading to talk about mutant IRES, when the changes are in the ORF.*

We accept this criticism and have made the appropriate changes in the Results section and in [Supplementary-material SD1-data].

*7) At the beginning of the Results part the authors should specify in which system they are working.*

The requested change has been made in revising the first sentence of Results.

*8) The green line is not mentioned in the legend to Figure 3.*

This criticism is addressed in the revised last sentence of the Figure 3 legend.

*9) A-site/E-site allostery has been recently reported by Ferguson et al., 2015, Mol Cell for human 80S ribosomes. This is relevant for the respective discussion in the third paragraph of the subsection “Rates of oligopeptide formation and Met-tRNA^Met^ cosedimentation”.*

We agree. We only became aware of the Ferguson et al. article (published November 2015) after we had submitted our initial MS to *eLife*. It is relevant to our manuscript with respect both to A-site:E-site allostery (see the response to Reviewer #2, point 3) and puromycin reactivity. The manuscript has been altered to include references to Ferguson et al. (2015) in the subsections: “Rates of oligopeptide formation and Met-tRNA^Met^ cosedimentation”, last paragraph; “Translocation of tetrapeptidyl-tRNA (Step 12) is much more rapid than of tripeptidyl-tRNA (Step 9), third paragraph and Discussion, fourth paragraph.

[Editors' note: further revisions were requested prior to acceptance, as described below.]

The manuscript has been improved but there are some remaining issues raised by Reviewer #3 that need to be addressed before acceptance. In considering these new comments, Reviewer #2 also asks that you carefully review the Materials and methods to insure that they contain all of the critical information that would be required to duplicate the work.

Reviewer 2 raises no specific objections to the previous submission (Revision 1) but we note that some experimental detail has been added to the Figure 3 legend in response to Reviewer 3 that might address the concern expressed in your overview relevant to “critical information.” Our point-by-point response to Reviewer 3 is given below.

*Reviewer #3:*

*In the revised version Zhang and colleagues have addressed many points raised by the reviewers and have improved their paper. However, there are still some points where discussion should be extended.*

*1) The authors now confirm that the effect of eEF2 is to inhibit k-1 and write that this is "consistent with its role as a translocase". However, the accepted role of EF-G/eEF2 is to accelerate translocation, not to inhibit back-translocation. This warrants additional discussion.*

We strongly disagree with the reviewer on this point. In fact, a principal role of EF-G, the prokaryotic equivalent of eEF2, has long been considered to be to inhibit back-translocation (Savelsbergh et al., Mol Cell. 2003 11:1517-23; Ratje et al., Nature. 2010 468:713-6), a view backed up by recent experiments. A reference to a particularly compelling set of results (Adio et al., 2015) has been added to the text to make this point explicit (subsection “Rates of Phe-TC binding to the 80S·IRES complex: Steps 1-3, structures 1 – 4”, end of second paragraph).

*2) The authors have pre-incubated 80S-FVKM-IRES complexes with eEF2-GTP for 1 – 2 hr. How they can rule out that GTP consumption has an impact on the experiment? Is it possible that accumulation of eEF2-GDP during pre-incubation results in a significant fraction of 80S FVKM-IRES-eEF2 complexes (between Structures 1 and 2) to slow down k2.*

We understand the reviewer’s concern, and have added additional text to make it clearer that we have adequately addressed this concern. As stated in the experimental section (subsection “Kinetic Experiments”) “Unless otherwise noted, all reactions were performed at 37 ^o^C in buffer 4 supplemented with 1 mM GTP”. To eliminate any ambiguity, we have made two additions to the Figure 3 legend. First, we added the phrase “containing 1 mM GTP”. Second, we added the sentence “eEF2 displays virtually no GTPase activity when it is not bound to the ribosome (Nygård and Nilsson, 1989)”. This makes it clear to the reader that the eEF2·GTP added as part of the Phe-TC solution will not undergo hydrolysis prior to its rapid mixing with 80S·IRES. Lastly, as mentioned in the text (Discussion, third paragraph), GTP is required for tight binding of eEF2 to the ribosome, so that eEF2·GTP should easily outcompete eEF2·GDP generated during the preincubation step for binding to 80S·IRES.

*3) The authors write that the anisotropy assay in Figure 3 reports on formation of Structure 3 from 1. However, as the co-sedimentation assay reports, during the measurement time there should be also formation of Structure 4 at later time.*

The reviewer is correct, but as Phe-TC is bound to the 80S·IRES complex in both Structures 3 and 4, any anisotropy change accompanying Structure 4 formation from Structure 3 is expected to be minor.

*4) The authors state that A-site-bound Phe-tRNA^Phe^ is labile. Can they estimate the off-rate k-2. Is it valid to neglect k-2 for numerical integration?*

Productive TC binding is accompanied by GTP hydrolysis, an essentially irreversible step. Thus, the lability of bound Phe-tRNA^Phe^ would not be measured by k_-2_, but by a separate reaction following dissociation of eEF1A·GDP from the ribosome, the kinetics of which we have not measured. However, this question by the reviewer prompted us to add a second sentence to the Figure 2 legend, explaining that Scheme 1 is simplified and neglects many substeps in the elongation process. As to the validity of neglecting the actual k_-2_, which describes the reversible binding of ternary complex, the answer is that it is always valid to use the minimum number of parameters to fit the available data. This results in obtaining *apparent* kinetic constants, as listed in Table 1. Different experimental results, which are outside the scope of this manuscript, could allow k_-2_ to be estimated, with, perhaps, altered apparent values of k_1_, k_-1_, and k_2_.